# Structural Insights into the Protein Mannosyltransferase from *Mycobacterium tuberculosis* reveal a WW-Domain-Like Protein Motif in Bacteria

Nicolas Géraud [1,7], Chloé Rivière[1,7], Camille Falcou [2], Gianluca Cioci[3,4], Carine Froment[1,5], Virginie Gervais[6], Julien Marcoux [1,5], Martine Gilleron [1], Jérôme Nigou [1], Emeline Fabre [1,8] & Michel Rivière [1,8] ✉

We have previously demonstrated that protein-O-mannosylation (POM), a widespread post-translational glycosyl modification of proteins, is a key virulence factor of *Mycobacterium tuberculosis* (Mtb), the world's deadliest infectious agent. Here, we report a detailed analysis of the structure-function relationship of MtPMT, the enzyme that catalyzes POM in Mtb. Using mutagenesis and in cellulo monitoring of POM activity, we demonstrate that, despite notable structural differences, MtPMT shares functional homologies with yeasts' PMTs in the mechanism of the sugar transfer from lipidic donors. Furthermore, we provide evidence that the selectivity for proline-rich target glycosylation sites that differentiates MtPMT from its eukaryotic homologues, relies on a WW-like domain, which preferentially interacts with proline-rich acceptor substrate analogues. This first identification of a functional WW-like domain in a prokaryotic protein raises questions about its potential evolutionary linkage with eukaryotic WW modules and provides new insights into PMT's acceptor-substrate recognition mechanism paving the way for the development selective inhibitors of MtPMT with potential therapeutic application against tuberculosis.

Protein-O-mannosylation (POM) is an essential post-translational glycosyl modification of proteins, conserved across species from yeast to animals[1,2]. In higher eukaryotes (including, Drosophila, zebrafish, mice, rats, and humans), defects of POM cause severe developmental, neurological and muscular disorders, leading to embryonic death and congenital defects. Long thought to be restricted to eukaryotes, POM has been shown to occur in actinobacteria[3–5] including the major human pathogen *Mycobacterium tuberculosis* (Mtb) responsible for the world's deadliest infectious disease: tuberculosis (TB)[6–8]. We have previously reported that O-mannosylation of Mtb's proteins is essential for Mtb infectivity, multiplication and virulence in the host[9,10].

In Mtb, POM is catalyzed by a single enzyme, MtPMT, coded by the *rv1002c* gene[5,9]. While several MtPMT target mannoproteins have been characterized, none fully account for the attenuated virulence observed in the MtPMT knockout mutant. Thus, broad blockade of the POM process through pharmacological inhibition of MtPMT constitutes an attractive target[11,12] that could open the way to the development of a "non-traditional" anti-virulence strategy[13–15] to complement conventional treatments against TB[8,16,17]. However, validation of POM as a viable strategy to reduce Mtb infectivity remains challenging due to the lack of known MtPMT inhibitors and limited understanding of the enzyme's molecular mechanisms, including its activity and target selectivity in bacteria. To address these

[1]Institut de Pharmacologie et de Biologie Structurale, IPBS, Université de Toulouse, CNRS, Université Toulouse III, Toulouse, France. [2]Architecture et Fonction des Macromolécules, UMR7257 CNRS - Aix-Marseille University, Marseille, France. [3]Toulouse Biotechnology Institute (TBI), Université de Toulouse, CNRS, INRAE, INSA, Toulouse, France. [4]Plateforme Intégrée de Criblage de Toulouse, IPBS, Université de Toulouse, CNRS, UPS, Université de Toulouse III, Toulouse, France. [5]Infrastructure nationale de protéomique, ProFI, FR 2048 Toulouse, France. [6]Université Paris-Saclay, CEA, CNRS - Institute for Integrative Biology of the Cell (I2BC), Gif-sur-Yvette, France. [7]These authors contributed equally: Nicolas Géraud, Chloé Rivière. [8]These authors jointly supervised this work: Emeline Fabre, Michel Rivière. ✉e-mail: Michel.Riviere@ipbs.fr

**Fig. 1 | Role and conformational model of the** *Mycobacterium tuberculosis* **"dolichyl-phosphate-mannose-protein-*O*-mannosyltransferase"** (#UniProt: P9WN05). **a:** Similar to eukaryotes, *M. tuberculosis* protein-*O*-mannosylation is carried out at the membrane by a protein-*O*-mannosyl-transferase, MtPMT, which transfers a mannosyl residue from a polyprenol-phosphate-mannose to a serine or a threonine on a nascent protein being translocated by the universal Sec protein secretion system. In Mtb, the primary mannose can be further elongated by sequential addition of mannosyl units by other glycosyltransferases, including the PimE mannosyl transferase[1]. **b:** AlphaFold 3 ribbon conformational model of *M. tuberculosis* MtPMT (MtPMT[AF3]) colored in a rainbow sequence outlining the topological patterns addressed in the present paper (EL: external loop; HH: horizontal helix; TMH: transmembrane helix, see figure S1 for MtPMT[AF3] model quality metrics). **c:** Superposition of the model and the structures of the *Saccharomyces cerevisiae* protein-*O*-mannosyl transferases ScPMT1 (top, in light pink, RMSD = 1.956) and ScPMT2 (bottom, in beige, RMSD = 1.721) resolved by Cryo-EM (PDB 6P2R), stressing the similarity between bacterial and eukaryotic PMTs. The MIR domain, specific to eukaryotes is indicated. **d:** Rainbow-colored topological sketch of MtPMT embedded in the bacterial membrane showing the eleven TMHs and the conserved amino-acids (circles, red: acidic, blue: basic and green: polar residue) presumed to be involved in the mannose transfer reaction. (Nt: N-terminus, Ct: C-terminus).

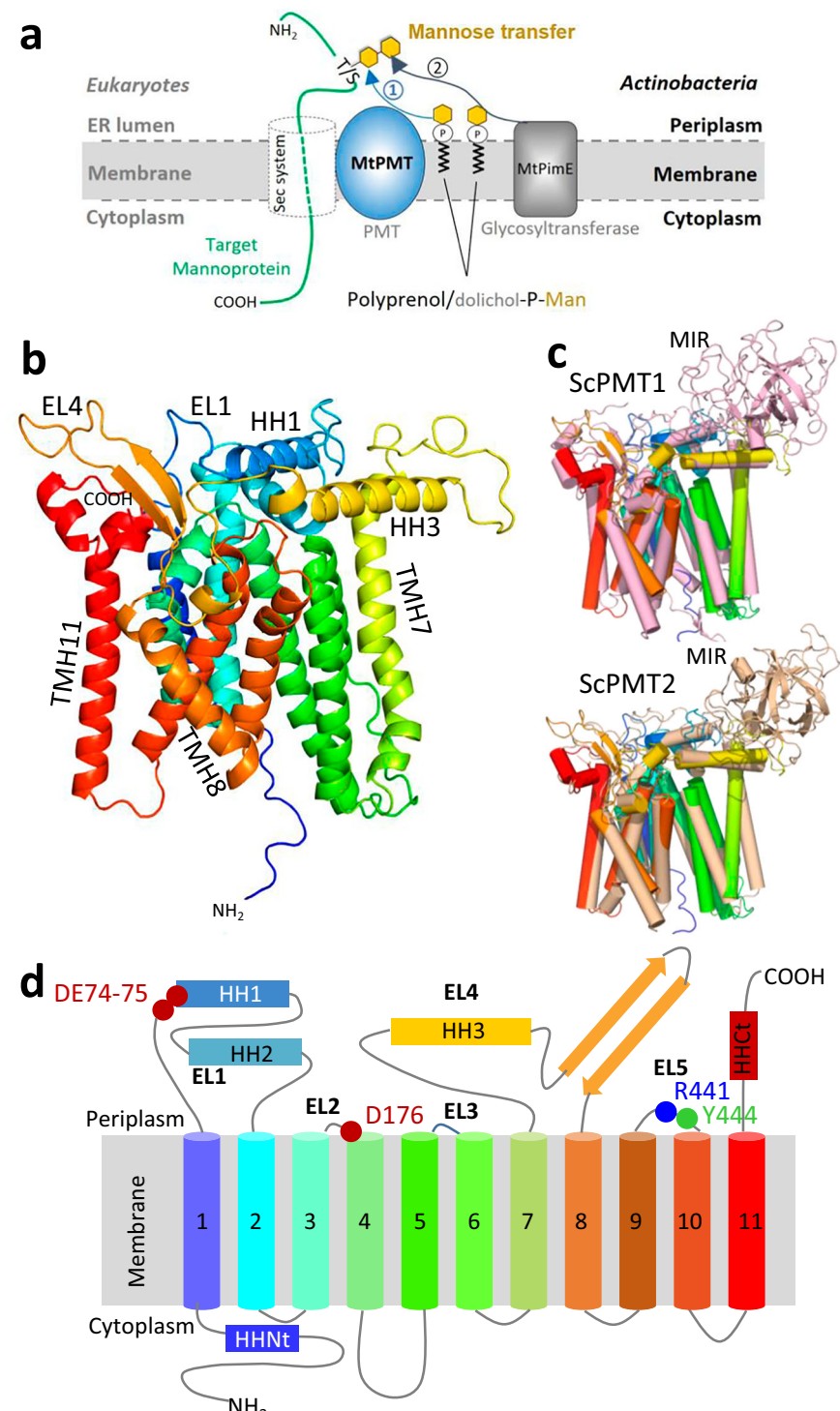

challenges, we investigated the structure-function relationships of MtPMT, drawing upon insights from homologous eukaryotic enzymes.

In eukaryotes, POM localizes to the lumen of the endoplasmic reticulum. This process is catalyzed by specific membrane-associated glycosyltransferases[2] named protein-*O*-mannosyl transferases (EC2.4.1.109) including PMTs (denoted POMT in animals). These enzymes transfer a mannosyl residue from a phospho-isoprenoid donor to the hydroxyl group of a threonine or serine on a nascent acceptor protein (Fig. 1a)[18–20]. This initial mannose can serve as an acceptor substrate for complementary glycosyltransferases that elongate the oligosaccharide by processive addition of sugars[9,18].

PMTs are part of the 'non-Leloir' C-type glycosyltransferase (GT-Cs) superfamily, which includes several essential membrane-bound enzymes involved in diverse glycosylation reactions such as N-glycosylation, glypiation, arabinosylation, O-mannosylation, or C-mannosylation. A defining hallmark of GT-Cs is their ability to utilize lipid-phosphates as sugar donors regardless of the acceptor substrates—proteins, lipids, or carbohydrates—that they glycosylate[21–23]. Another key feature of the members of this superfamily is their polytopic architecture, typically consisting of 11 to 16 transmembrane helices (TMHs) that form a structural framework. These hydrophobic TMHs are interspersed with a variable number of soluble loops protruding into the cytosol or into the lumen of the endoplasmic

reticulum[22,24-26]. While slight variations in folding patterns have been observed[22,27,28], this overall architecture is predominant in the limited number of GT-C structures that have been solved, including in the reported structure of the naturally occurring heterodimeric complex ScPMT1/ ScPMT2 of *S. cerevi*siae[29]. Indeed, this unique structure of PMT so far available confirmed that each PMT protomer of the complex adopts such a characteristic configuration with 11 TMHs and a bulky luminal β-trefoil fold corresponding to the MIR domains[29]. This cryo-EM resolved structure also provided key insights into the position of the alkyl chain of the donor substrate analogue between TMHs 6 and 7 of ScPMT1, as well as the conserved arrangement of several strategic amino acids that may assist in positioning and activating the hydroxyl acceptor for mannose transfer. However, while a trace of the PYT acceptor peptide analogue was detected in the luminal pocket of ScPMT2, the molecular details of the enzyme inter-action with the acceptor substrate remain elusive[29]. These structural insights have enhanced our understanding of the molecular mechanisms underlying mannose transfer and donor substrate recognition in eukaryotic PMTs. Nonetheless, the factors governing the selective recognition of serine/ threonine (S/T)-containing peptides as acceptor substrates for mannosylation are still poorly understood.

In bacteria, the structure-function relationships of PMTs remain lar-gely understudied and elusive. While analogies with GT-Cs, including yeast PMTs, exist, bacterial homologs have significant genetic, functional and structural divergences[30]. For instance, PMTs are ubiquitous in animals and fungi, whereas, until now, bacterial protein-*O*-mannosylation seems to be restricted to the only members of the phylum Actinomycetota. Further-more, the bacterial genomes analyzed so far, possess a single PMT-encoding gene unlike opisthokonts which express from 2 to 6 several non-redundant PMT orthologs. In addition, while POM is essential in eukaryotes, it has been found totally dispensable for bacterial physiology under laboratory growth conditions[4,9,31].

Functionally, yeast PMTs are localized to the endoplasmic reticulum membrane, where they associate with the Sec63 translocon complex to glycosylate nascent proteins during translocation (Fig. 1a)[25]. In contrast, consistent with the absence of ER membranes in prokaryotes, the POM process in mycobacteria has been associated with the plasma membrane or the cell envelope fraction[32]. Likewise, the dependence of the POM pathway on the general bacterial protein secretion pathway (Sec pathway)[5] also supports the localization of MtPMT to the mycobacterial plasma mem-brane, although this localization has yet to be formally confirmed.

Finally, the structure of bacterial PMTs differs markedly from their eukaryotic counterparts. Notably, bacterial PMTs lack a large fragment in their external loop 4 (EL4), corresponding to the bulky soluble MIR domains. In eukaryotes, these MIR domains have recently been reported to contribute to PMTs heterodimerisation and to the processive displacement of the mannosylated acceptor substrate[33]. The absence of MIR domains in bacterial PMTs suggests that the bacterial enzymes, which may represent ancestral forms of evolved eukaryotic PMTs[30], likely operate through dis-tinct molecular mechanisms.

Given these substantial differences, we conducted a detailed investi-gation into the structure-function relationships of MtPMT to gain deeper insight into the specific molecular mechanisms of bacterial PMTs. Our primary objectives were to assess the topological and functional similarities between MtPMT and its eukaryotic counterparts and to explore the potential druggability of the bacterial enzyme in vivo. Furthermore, in the search for distinctive features that could be exploited to selectively target bacterial MtPMT, we identified a striking proline-rich composition in the mannosylated peptides of *M. tuberculosis*. This finding suggests a strong preference of MtPMT for proline-rich target peptides, prompting further exploration of the enzyme's mechanism for acceptor substrate recognition. Functional mutagenesis and in vitro interaction studies provided additional insights, revealing an unexpected role for the C-terminal domain of EL4 in the selective recognition of proline-containing peptides, possibly through a mechanism resembling that of eukaryotic WW domains. These original results constitute a significant step forward in understanding the

structure-function relationships of MtPMT, paving the way for the rational design of inhibitors specifically targeting Mtb protein O-mannosylation.

## Results
### Predictive model of MtPMT topology

To support the structure-function analysis of MtPMT, we generated high-confidence 3D structural models of the bacterial enzyme using various web-based protein 3D structure prediction tools: Phyre2[34], Robetta[35], I-TASSER[36], and AlphaFold[37]. The four programs predicted very similar conformations with pairwise average RMSD values ranging from 1.1 to 2.3 over more than 2000 atoms (Fig. 1b). We also observed remarkable alignment coverages between the predicted models (Fig. S1) and the recently resolved cryo-EM structures of the ScPMT1/ ScPMT2 heterodimer[29] (PDB 6P2R) (Fig. 1c). These yeast orthologues respectively share 26% and 23% sequence identity with MtPMT (Fig. S2). In addition, these pairwise sequence homologies, particularly strong in the hydrophobic domains, are consistent with a model based on the presence of 11 transmembrane helices (TMHs) (Fig. 1d)[5]. This arrangement represents the characteristic modular architecture, featuring the canonical InterPro database modules PMT (PF02366) and PMT_4TMC (PF16192), which define the prototypical core of 11 transmembrane helices (TMHs) observed in PMT family members and most resolved GT-C structures to date (Figs. S3 and S4)[22]. It is worth noting that the predictive models inferred from the MtPMT primary sequence clearly highlight its relatively low hydrophilicity compared to most GT-C structures. Indeed, the absence of the soluble MIR domains found in eukaryotic PMTs results in a much higher average hydropathic balance (GRAVY MtPMT + 0.431 ver-sus −0.002 and −0.052 for ScPMT1 and ScPMT2) (Fig. S5). The difference in solubility compared to the eukaryotic homologs makes MtPMT isolation highly challenging and led us to adopt genetic approaches for structure-function analyses.

### Experimental assessment of the in situ functional topology of MtPMT

The odd number of TMHs predicted by molecular models, suggested that MtPMT may adopt a "trans" membrane configuration with its extremities located on opposite faces of the bacterial plasma membrane, similar to the orientation of eukaryotic PMTs in the ER membrane (Fig. 1a). To confirm this topology and to determine the functional arrangement of the bacterial enzyme directly, MtPMT-reporter chimera proteins were ectopically expressed in *M. smegmatis* lacking its endogenous PMT gene MSMEG_5447 (*M. smegmatis* ΔMsPMT)[9] (Fig. 2a). The constructs were designed to reveal the subcellular localization of the PMT reporter termini. High-resolution imaging of live bacteria expressing the cytoplasmic reporter mCherry fluorescent protein fused to the MtPMT N-terminus (mCherry-NterMtPMT)[38] revealed a homogeneous distribution of fluores-cence around the cell, consistent with a peripheral localization of MtPMT in the bacterial envelope (Fig. 2b). Colocalization of the mCherry-NterMtPMT fusion with the fluorescent membrane intercalating dye PKH67[39], further validated the integration of MtPMT into the bacterial plasma membrane. In contrast, cells expressing the reverse fusion with the mCherry protein at the Cter extremity of the MtPMT sequence (MtPMTCter-mCherry) showed much lower fluorescence intensity, hindering high-resolution imaging (Figs. S6 and S7). The lower fluorescence observed for this latter strain expressing the MtPMTCter-mCherry construct is consistent with the C-terminal fluorescent probe being exposed to the highly oxidative peri-plasmic environment, which is detrimental to the correct folding of secreted fluorescent proteins[40].

To confirm the extra-cytoplasmic localization of the Cter end of MtPMT in situ, we fused the enzyme to the *E. coli* periplasmic alkaline phosphatase PhoA (Fig. 2a). Because it is only active when secreted, the PhoA tag is commonly used as a reporter of protein export in mycobacteria[41,42]. Its periplasmic activity is easily detectable in the presence of a chromo-genic substrate on solid culture medium. This was clearly illustrated by the blue color of the positive control expressing the MmpS4 protein fused to PhoA (MmpS4Cter-PhoA), a well-known reporter protein secreted by

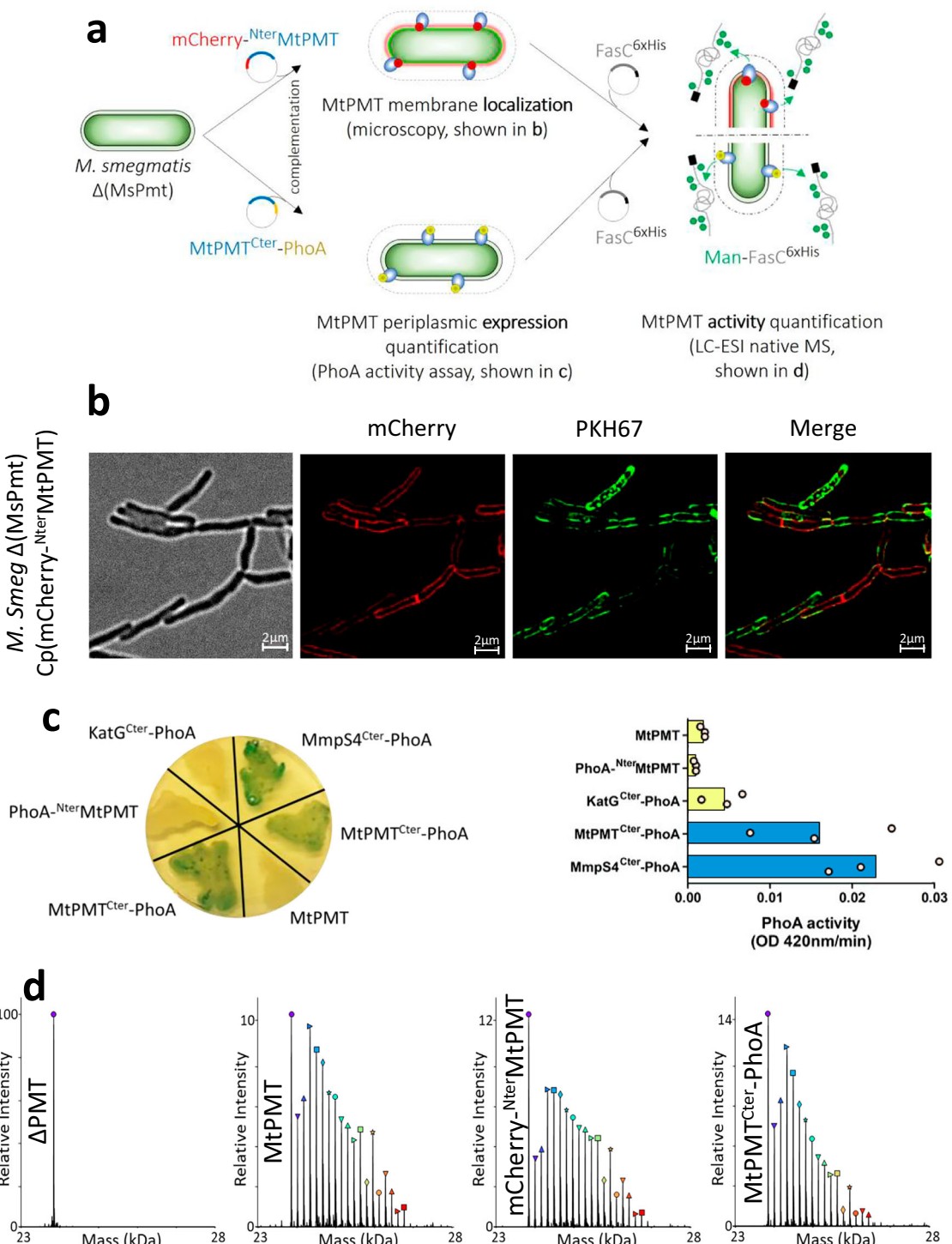

**Fig. 2 | In-situ evidence for membrane insertion and functional topology of MtPMT. a** Diagram of experimental strategy: *M. smegmatis* Δ(MsPmt) was complemented with plasmids encoding chimeric subcellular compartment reporter proteins mCherry-[Nter]MtPMT or MtPMT[Cter]-PhoA. Ectopic expression and localization of the reporter tag were monitored by microscopy or quantification of PhoA activity. To test the functionality of MtPMT, an *M. smegmatis* Δ(MsPmt) strain expressing the wild type or hybrid MtPMT proteins was secondarily transformed with a plasmid encoding the MtPMT acceptor substrate protein FasC[His]. MtPMT activity was monitored by LC-ESI-MS detection of the mannosylation patterns of native FasC[His]. **b** Transmission and high-resolution fluorescence microscopy of *M. smegmatis* Δ(MsPmt) bacteria expressing the mCherry-[Nter]MtPMT fusion protein (red) and stained with the membrane marker PKH67 (green), showing the peripheral location of the MtPMT fusion protein.

(Scale bars: 2 μm). (Cp: complemented) **c** Periplasmic PhoA activity of *M. smegmatis* Δ(MsPmt) bacteria complemented with MmpS4[Cter]-PhoA (periplasmic control), MtPMT[Cter]-PhoA, KatG[Cter]-PhoA (cytoplasmic control), PhoA-[Nter]MtPMT or untagged MtPMT. PhoA activity was detected on a solid support (left) using a chromogenic substrate and quantified in the periplasmic fractions by following pNPP hydrolysis at 420 nm (right). PhoA activity was quantified on three independent clones of each strain). **d** Intact protein high-resolution MS deconvoluted spectra showing the mannosylation patterns of native FasC[His] purified by metal affinity column from the culture supernatants of *M. smegmatis* Δ(MsPmt) complemented with a mock plasmid (ΔPMT) or with plasmids encoding either the wild type MtPMT or the mCherry-[Nter]MtPMT or MtPMT[Cter]-PhoA fusion proteins (peaks relative intensity and symbol legend are reported in Table S1).

mycobacteria[43,44] (Fig. 2c). On the other hand, as expected, we did not detect phosphatase activity in the negative control strain, which express PhoA fused to the intracellular catalase-peroxidase KatG (KatG[Cter]-PhoA)[44]. Similarly, the complementary negative control strain expressing wild-type MtPMT remained uncolored, confirming the absence of interfering endogenous phosphatase activity. Finally, the blue staining of the recombinant strain colonies expressing the MtPMT[Cter]-PhoA fusion protein attested to the periplasmic localization of PhoA when placed at the C-terminus of MtPMT (Fig. 2c). In contrast, strains expressing the inverse PhoA-[Nter]MtPMT fusion protein were unable to hydrolyze the chromogenic substrate, consistent with PhoA cytosolic localization when fused to the MtPMT N-terminus (Fig. 2c).

To confirm that this orientation corresponds to the functional configuration of the enzyme in the membrane, we controlled the mannose transferase activity of these reporter hybrid proteins. With this aim, cells expressing the mCherry-[Nter]MtPMT or MtPMT[Cter]-PhoA proteins were transformed with a plasmid coding for the *M. smegmatis* hexa-histidine tagged FasC[His], an acceptor substrate for mannosylation by MtPMT (Fig. 2a)[9]. The mannosylation profile of the secreted FasC[His] was then monitored by intact protein high-resolution mass spectrometry (HR-MS) after metal affinity chromatography[45]. As expected, FasC[His] protein purified from the culture medium of the ΔMsPMT strain showed a single proteoform at 23,845 Da corresponding to the non-mannosylated protein (averaged $MW_{Calc}$ 23,847Da) (Fig. 2d). Absence of additional glycoforms confirmed the lack of "protein mannosyl transferase" activity in the MsPMT knock-out strain. In contrast, the series of peaks separated by multiples of 162 Da, observed in the deconvoluted HR-MS spectrum of FasC purified from the strain expressing MtPMT (ΔMsPMT:MtPMT), indicated the presence of multiple glycoforms differing in the number of hexose units. This characteristic profile, which evidences the presence of 18 FasC glycosylation states, confirms that MtPMT constitutively expressed in *M. smegmatis* is enzymatically active.

Interestingly, HR-MS analysis of secreted FasC[His] proteins purified from the culture media of strains expressing the mCherry-[Nter]MtPMT or MtPMT[Cter]-PhoA fusion proteins revealed multiple FasC glycoforms, as observed for the wild-type enzyme. This proved that terminal fusion of MtPMT with mCherry or PhoA does not significantly impact its mannosyl transferase activity. Overall, these observations indicate that MtPMT expressed in *M. smegmatis* adopts a trans-membrane functional configuration with its N-terminus immersed in the cytosolic compartment and its C-terminal domain protruding from the opposite side of the membrane in the bacterial periplasm.

## Functional validation of selected amino acids in the catalytic site
To further assess the functional homology of MtPMT with the eukaryotic enzymes, we next explored the impact of selected amino-acid substitutions on the capacity of MtPMT to mannosylate the FasC acceptor substrate. First, we tested the role of the highly conserved acidic aspartate D74, part of the DE or DxD motif, which is invariably located upstream of helix HH1 in the first external loop of most C-type glycosyl transferases (Figs. 1d and S3). To date, this amino acid is the only one that has been definitely shown to be essential for the catalytic activity of PMTs[5,21,46]. We therefore assessed the capacity of the MtPMT D74A mutant to mannosylate the secreted FasC[His] reporter protein using a Concanavalin A (ConA)-based enzyme-linked lectin assay (ELLA)[45]. Quantification of the amount of ConA bound to immuno-immobilized FasC[His] from crude supernatant revealed a complete lack of mannosylation of the FasC[His] secreted by bacteria expressing the D74A mutant (Fig. 3a). These data confirm that aspartic acid 74 is essential for MtPMT activity. Similarly, substitution of the acidic aspartate by an oppositely charged asparagine (D74N) also led to the suppression of FasC[His] mannosylation (Fig. 3a). Since our result diverged from the reported lack of effect of the D74N substitution on *S. coelicolor* PMT activity[47], we checked whether the absence of FasC[His] mannosylation resulted from lack of expression of this mutant. A control for expression of the chimeric D74N MtPMTC[ter]-PhoA mutant indicated that the lack of FasC mannosylation is

not due to defective gene expression or instability of the full-length enzyme, but rather to the loss of catalytic activity of the enzyme (Fig. 3b).

To gain further insight into the factors underlying the importance of D74, we mutated the aspartic acid to glutamic acid, a longer side chain. The D74E mutant only marginally restored enzyme activity, suggesting that the spatial position of the D74 carboxyl relative to substrates is as important as its charge. Finally, di-substitution of the $D_{74}E_{75}$ residues with the reversed doublet $E_{74}D_{75}$ also failed to restore FasC[His] mannosylation, revealing that the respective functions of these two acidic residues are not redundant or exchangeable for MtPMT enzymatic activity (Fig. 3a).

Alongside this pivotal D74, most glycoenzymes proceeding through inversion of the sugar anomeric configuration, such as oligosaccharyl transferases and β-glycosidases, contain a second highly conserved aspartic acid (often associated with an $Rx_{4-6}D$ or $Dx_{4-6}D$ motif) located on their predicted second luminal loop[21,48] (Figs. 1d and S3). This second aspartate plays an essential role in the catalytic mechanism of the oligosaccharyl transferases PglB[49,50] and STT3[51]. Independent studies suggested that it either acts as a general base assisting the transfer of sugar by nucleophilic substitution (Fig. 3e)[52] or as a chelator of divalent metal cation stabilizing the negatively charged phosphate of the leaving sugar lipidic donor[21]. Although the possible role of such an amino acid has not yet been reported in PMTs, a careful search in the spatial vicinity of the putative active site of MtPMT revealed the presence of D176 (Fig. 3c). Remarkably, this residue aligns with the conserved D189 and D203, respectively situated in the immediate neighborhood of the catalytic residues D77 and D92 in the published structure of the ScPMT1/PMT2 complex[29]. Therefore, we investigated whether this second conserved aspartic acid situated in EL2 contributes to MtPMT activity. Interestingly, substitution of D176 with a short-chain uncharged alanine almost completely abolished the mannosylation of FasC (Fig. 3a). Similarly, replacing the D176 carboxyl group with an asparagine carbamoyl group in the D176N mutant also resulted in the abrogation of FasC mannosylation (Fig. 3a). As above, we verified that the lack of POM activity of the D176N mutant was attributable to the loss of enzymatic activity rather than to a defect in expression of the mutated enzyme (Fig. 3b). In addition, substitution of the acidic aspartate with a glutamic acid, which carries a longer side chain, slightly restored mannosylation activity, as revealed by the much reduced but detectable level of ConA binding to the FasC secreted by the D176E mutant strain (Fig. 3a). Thus, the charge and the precise positioning of the aspartic acid carboxyl group relative to the catalytic center are both crucial for MtPMT activity.

Finally, closer inspection of the conformational model of MtPMT also highlighted two conserved amino acids, namely R441 and Y444, located spatially close to the catalytic D74 (Figs. 3c and S3) though distant in the primary sequence since they lie in the predicted fifth external loop (Figs. 1d and S3). It is noteworthy that this $Rx_2Y$ motif coincides perfectly with the respective homologous pairs R649-F652 and R671-Y674, localized inside the pockets hosting the sugar donor polar phosphate and the acceptor peptide at the lumenal face of the ScPMT1 and ScPMT2 structures (Fig. 3c)[29]. To verify whether this striking topological array, conserved in MtPMT, contributes to the activity of the bacterial enzyme, we thus analyzed the functionalities of these two residues. Alanine substitution of Y444 led to complete loss of FasC mannosylation, unambiguously demonstrating the functional significance of this residue for MtPMT activity (Fig. 3a). However, almost half of the FasC mannosylation level remained when an aromatic phenylalanine was substituted for Y444, as the one found at the corresponding position (F652) in ScPMT1 (Figs. 3a and S2)[29]. This observation highlights the importance of an aromatic ring at this position.

Similarly, mutations targeting asparagine R441 indicated that the side chain charge was crucial for MtPMT enzymatic activity, since changes into aliphatic alanine (R441A) or leucine (R441L) completely abolished FasC mannosylation (Fig. 3a). In contrast, the conservative substitution of R441 with a basic lysine (R441K) almost completely preserved enzymatic activity, as shown by the limited decrease in ConA binding level (Fig. 3a). Unlike the conservative D74E or D176E mutations, which drastically affected MtPMT enzymatic activity, the restoring effect of the R441K substitution confirms

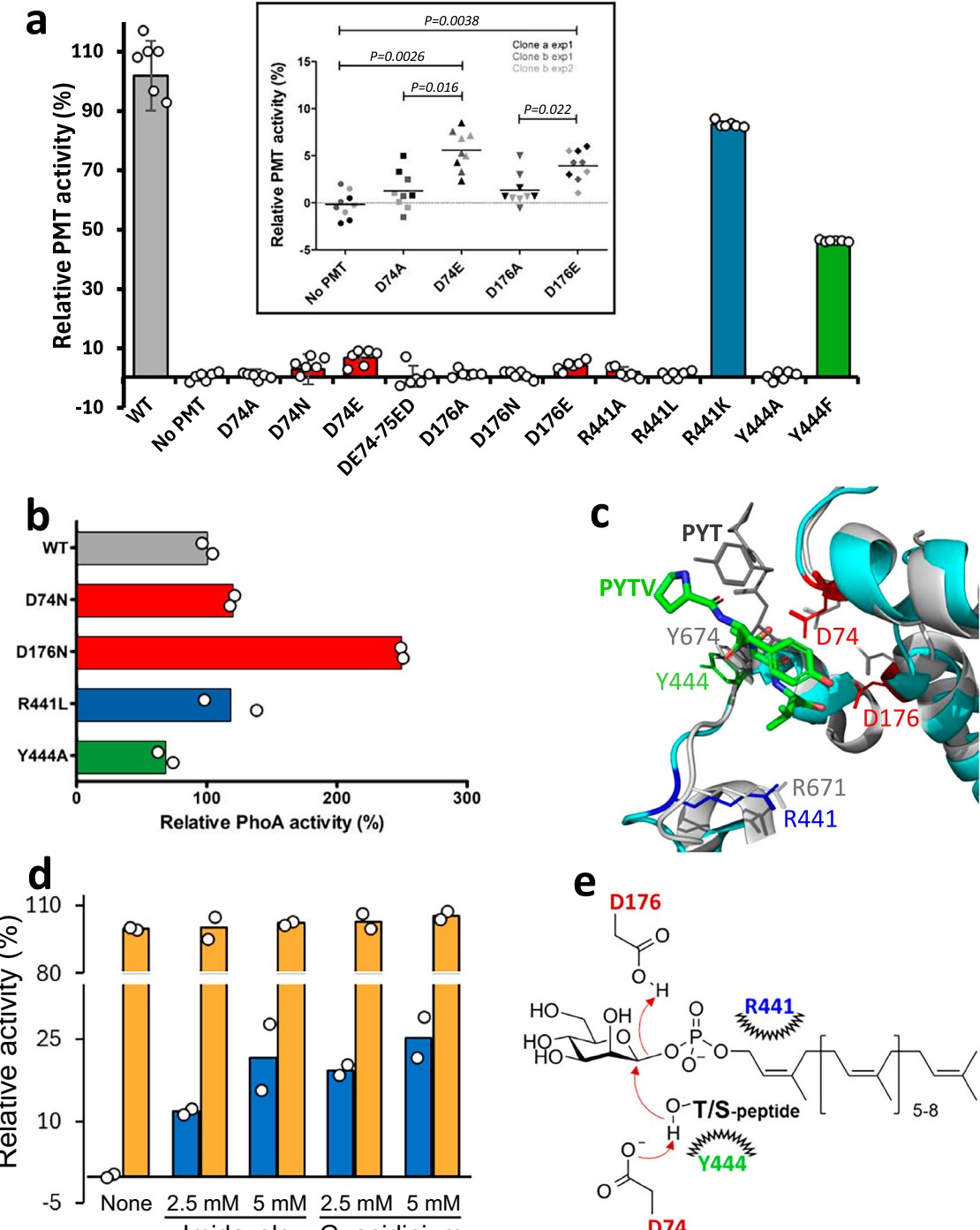

**Fig. 3 | Identification of MtPMT active-site residues essential for enzymatic activity reveals its potential as a druggable Mtb virulence factor. a** Site-directed mutagenesis highlights the functional importance of residues D74, D176, R441, and Y444 within the MtPMT catalytic center. (Bars are color-coded according to the scheme shown in panel c; Data represent the mean of biological replicates from 2 clones of each mutant strains and the error bars represent the standard error of the mean (s.e.m.). Inset graph data were analyzed using an unpaired Student's *t*-test (Pertinent and significant one-tailed *p* values *p* < 0.05 are indicated). **b** Periplasmic phosphatase activity assay confirms expression of inactive MtPMT mutant variants fused to a PhoA tag (as above, PhoA activity was quantified on two independent clones of each strain). **c** Close-up view of the AlphaFold 3 model of the PYTV peptide (shown as colored sticks) docked within the active site of MtPMT (cyan cartoon) (ipTM = 0.61, pTM = 0.92), superposed on the cryo-EM structure of ScPMT2 (gray

cartoon) bound to a PYT peptide (gray lines; PDB: 6P2R, Bai et al.). The structural alignment (PyMOL RMSD_MtPMT–PYTV / ScPMT2–PYT = 2.39 Å) suggests a slightly altered peptide orientation in MtPMT, yet the threonine hydroxyl of PYTV remains within hydrogen-bonding distance of the catalytic D74, supporting the plausibility of the modeled interaction. **d** In vivo partial rescue of enzymatic activity in a genetically inactivated MtPMT mutant (R441A, blue bars) by supplementation with guanidinium or imidazole, mimicking the arginine side chain. Wild-type MtPMT is shown in orange. This supports the feasibility of chemically modulating enzyme activity in situ with small exogenous chemicals. **e** Proposed catalytic mechanism of MtPMT involving a general acid/base direct displacement reaction. D74 acts as a base to activate the hydroxyl group of the threonine or serine acceptor, while D176 serves as a proton donor. Residues R441 and Y444 stabilize both the phosphate group of the lipid-P-mannose donor and the acceptor substrate.

that the positive character of the R side chain is more important than its relative spatial position.

Considering the critical role of the R441 guanidinium moiety for enzyme activity, we attempted to rescue the functionality of the corresponding inactive R441A mutated proteins by supplying live bacteria directly with small chemicals that have electronic and steric properties that complement the vacant molecular space of the arginine missing side chain[53]. Interestingly, we found that either guanidine chloride or imidazole were able to restore FasC mannosylation to ~25 ± 5% of the level of the wild type MtPMT[54] (Fig. 3d). On the other hand, attempts to similarly complement the D74A or D176A inactive mutants with sodium acetate[55] or sodium azide[54] were ineffective, consistent with the required precise position of the carboxyl side chains evidenced by the inactivity of the D74E and D176E substituted mutants. Thus, the efficient in vivo chemical rescue of the inactive R441A mutant suggests accessibility of the active site pockets and constitutes compelling evidence that MtPMT activity can be modulated by small chemicals *in cellulo*[56].

Overall, these results confirmed the role of D74, but more importantly provided definitive experimental evidence that the highly conserved D176, R441 and Y444 residues are essential for MtPMT activity in living bacteria (Fig. 3e). The detailed chemical roles that these residues play in PMT catalysis, the mannose transfer reaction, and/or interaction with the lipidic glycosyl-donor remain to be clarified[29]. Nevertheless, our observations confirm that the molecular determinants involved in the enzymatic reaction and in the interaction with the lipidic substrate are highly conserved among PMTs.

## MtPMT contains a proline-rich motif binding domain

Next, we looked for divergences between eukaryotic and bacterial PMTs, focusing in particular on the much less well-defined molecular basis of the selective recognition of the acceptor peptide substrate. To understand how S/T-containing peptides are selected as acceptor substrates for MtPMT-mediated mannosylation, we searched for potential tags that are shared by mycobacterial mannopeptides and that might be recognized by the enzyme. To this end, we aligned the primary sequences of all mannosylated peptides in Mtb that have been fully characterized by MS so far ($n = 48$; Supplementary data 1)[57]. Consistent with previous attempts, no consensus sequence for mannosylation emerged from this analysis. However, comparing the residues surrounding the unmannosylated versus the mannosylated S or T in Mtb mannoproteins unveiled a significant enrichment of a proline residue around the mannosylated S/T[58,59] (Fig. 4a). According to the AlphaFold structure predictions of the corresponding Mtb mannoproteins, these (P rich [S/T]) sequences generally lie in poorly structured N or C terminal segments of the protein and may form random coils[60]. Such unstructured organization of glycosylation site-containing sequences suggests the selective detection of the torsion angle constraints of the ligand backbone induced by the rigidity of the proline pyrrole ring.

The search for homologies with non-glycosyl transferase proteins that might share the ability of MtPMT to interact with similar primary sequences prompted us to focus on proteins recognizing proline-containing O-gly-copeptides or proline-rich motifs (PRMs). Disappointingly, the conformation of MtPMT did not present any obvious match to the canonical scaffolds of any of the six currently recognized families of PRM-binding proteins[61–63] or of a known "proline-specific O-glycopeptidase"[64]. However, a closer look at the topology of the MtPMT periplasmic face surrounding the catalytic pocket defined by the four conserved amino acids D74, D176, R441, and Y444 revealed a high density of aromatic amino acids lining the bottom of the pocket (Fig. 4b, c). This arrangement, comprising the six aromatic residues, F73, H353, W355, Y371, Y444, and F445, strongly resembles the characteristic "aromatic cradle" found in all PRM-binding domain families. It has been shown that the relative spatial disposition of the exposed aromatic side chains of these highly conserved hydrophobic clusters contributes to proline-rich peptide selection through ring stacking involving π-π interactions[62–65]. Interestingly, the importance of the aromatic

character of Y444 for MtPMT activity is consistent with such a role and supports the analogy of this particular arrangement with the "aromatic cradle".

In addition, it should be noted that the singular β-sheet hairpin-like structure adopted by the EL4's C-terminal (EL4Cter) segment P355–W399 of MtPMT faces the D74 catalytic residue of EL1 on the opposite side of the catalytic pocket. Together, EL1, EL4Cter and EL5 form a groove that may accommodate the acceptor peptide and guide its positioning down to the buried catalytic site in the protein core at the periplasmic face of the enzyme (Fig. 4b). Further searches for similarity identified significant homologies both in the primary sequence and in the secondary structure of the bacterial EL4Cter segment and the WW PRM-binding domains of several proteins (Fig. 4d)[66]. Indeed, the motif $[(P^{359}-x_2-W^{362})-x_8-(Y^{371})-x_{20}-(L^{392}-x_3-P^{396})]$ found in the EL4Cter domain shares 4 of the 5 amino acids that make up the characteristic signature of most WW modules reported to date $[(P-x_{0-3}-W)-x_{10-12}-(Y/F)-x_{12-15}-(W-x_{1-3}-P)]$ (Fig. S8)[67–70]. Moreover, the resemblance between the two antiparallel β-strands of the EL4Cter domain and the characteristic β-sheet secondary structure of the Pin1 WW module (Fig. 4e) further strengthens the analogy. Altogether, these observations led us to hypothesize that the periplasmic EL4Cter module may be crucial for the recognition and accommodation of the acceptor peptide ligand[63,65].

## Genetic assessment of the function of the MtPMT EL4 C-terminal domain

To support this hypothesis, we probed in vivo the functionality of this conformational motif by mutagenesis and functional analysis. Primary sequence alignments of MtPMT EL4Cter with the corresponding domains of yeast or human PMTs revealed low conservation with major divergences, notably due to a highly conserved insertion of nine additional amino acids between the two β-strands in mycobacterial PMTs (Fig. 5a–c).

Thus, we focused primarily on the two cysteines, C381 and C386, localized in this mycobacterial-specific insertion joining the two predicted β-strands. Indeed, exposure of this Cx$_4$C motif to the periplasmic oxidative environment can promote intramolecular disulfide bridge strengthening the hairpin-like structure (Fig. S9). However, substitutions of these cysteines, individually or concomitantly, with alanine or serine residues had little effect on the level of FasC glycosylation, as evidenced by the normal binding level of ConA to FasC (Fig. 5d). Thus, although the redox status of these cysteines remains undefined in-situ, it is clear that intramolecular disulfide bridge formation[71] between them is not necessary for enzyme activity.

We next probed the potential functional contribution of selected amino acids specifically conserved in mycobacteria, namely M364, S365, L366, and I373 (Figs. 5c and S9). Substitutions of these residues with alanine or amino acids carrying chemically equivalent (S365T) or completely divergent (I373Q) side chains only partially impacted FasC mannosylation, as deduced from the still significant level of relative ConA binding (Fig. 5d). The limited decrease in apparent FasC mannosylation induced by point-mutations proves that these mycobacteria-specific amino acids are not essential for MtPMT activity and suggests that they may contribute only partially to the function of EL4Cter.

On the other hand, we observed that mutations of amino acids conserved from bacterial to eukaryotic EL4Cter domains affected more drastically mannosylation of the reporter protein FasC. Among these, four positions corresponding to amino acids W355, W362, L392, and P396, located on either side of the hairpin (Figs. 5c and S9), were decisive for enzyme activity. As shown in Fig. 5d, the W355A mutation resulted in a 70% drop in the apparent level of FasC mannosylation. As above, quantification of the periplasmic PhoA activity of the recombinant mutated PhoA-fusion protein (W355A- MtPMT$^{Cter}$-PhoA) confirmed that the W355A mutation did not impair the expression of the enzyme but affected the process of mannosyl transfer to target glycosylation sites (Fig. 5e). The functional importance of an aromatic residue at this position was supported by the occurrence of a F or a Y occupying the equivalent situation in HsPOMT2 or ScPMT1 and HsPOMT1, respectively. However, surprisingly, conservative

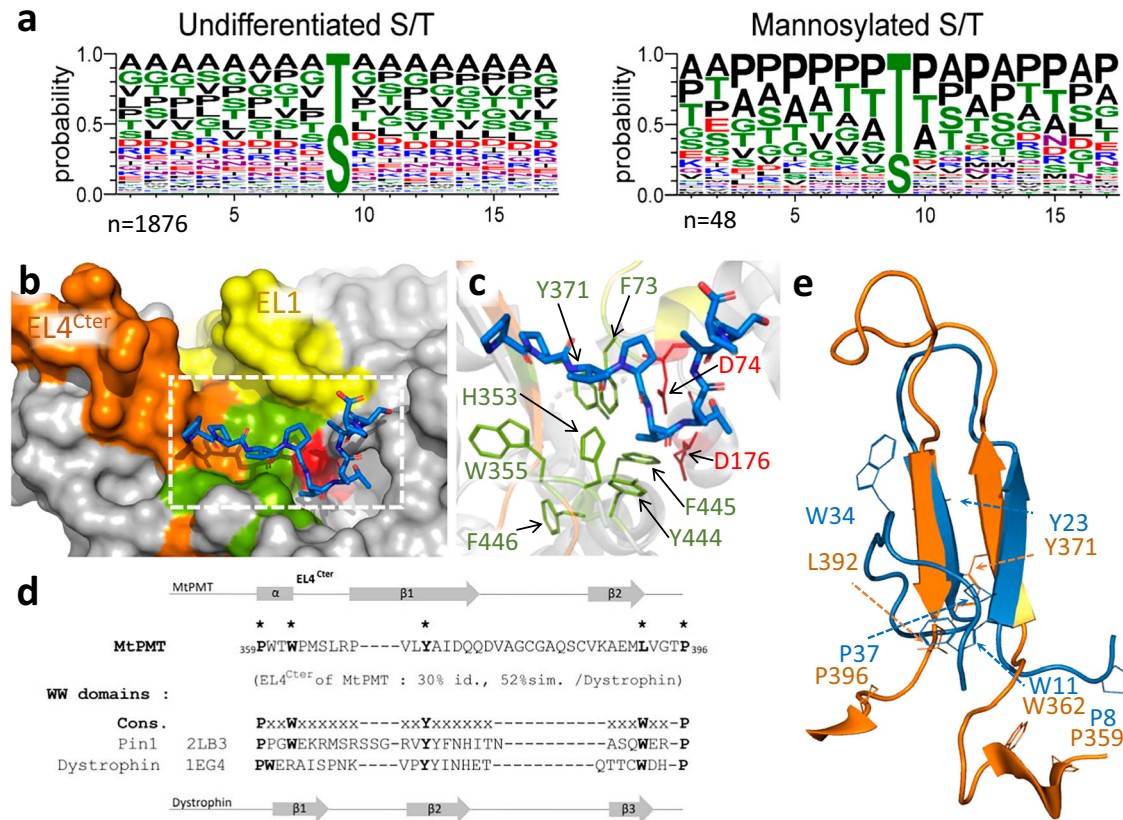

**Fig. 4 | Evidence supporting the role of MtPMT EL4Cter in the selective recognition of [S/T]-containing proline-rich peptides for mannosylation. a** Sequence analysis reveals preferential mannosylation of [S/T]-containing proline-rich peptides. The amino acid environment of all serine/threonine (S/T) residues in a set of experimentally characterized Mtb mannoproteins ($n = 44$)[3,4] (left panel; 17-mer peptides, $n = 1876$) was compared to that of experimentally validated glycosylation sites in the same proteins (right panel; 17-mer peptides, $n = 48$). Glycosylated S/T sites show significant enrichment in proline (P), alanine (A), and additional S/T residues surrounding the modification site (see Supplementary data 1 for full list). **b** Surface representation of the periplasmic face of the MtPMT$^{AF3}$ complexed with the Nter truncated Apa1 decapeptide [PPVPTTAASP] (ipTM = 0.69, pTM = 0.88), illustrating the substrate-binding groove. The [PPVPTTAASP] peptide is represented with carbon and nitrogen atoms in blue and oxygen in red. Key structural elements are highlighted: EL1Nter (yellow), EL4Cter (orange), the aromatic cradle (green), and the catalytic residue D74 (red). **c** Cartoon representation of the putative acceptor peptide binding site in MtPMT, highlighting residues that form the aromatic cradle. Coloring is consistent with (**b**). **d** Sequence alignment of the MtPMT EL4Cter domain with the WW domains of human Pin1 and dystrophin, showing 52% sequence similarity. The WW domain consensus (Cons.) is shown with five conserved residues marked in bold and indicated by asterisks; four of these are conserved in MtPMT. Notably, the second tryptophan (W), typically located in the β3 strand of canonical WW domains, is replaced by leucine (L) in MtPMT EL4Cter. Secondary structural elements (α-helices and β-strands) are indicated by rectangles and arrows, respectively. **e** Alignment of the MtPMT EL4Cter domain (orange) from the AlphaFold 3 model with the WW domain of human Pin1 (PDB: 1PIN, blue), revealing conformational homology and suggesting a shared mode of peptide recognition mechanism.

substitution of the MtPMT W355 with an aromatic phenylalanine completely abolished ConA binding to the secreted FasC, indicating a total loss of function of the enzyme upon W355F substitution (Fig. 5d). This observation shows that the tryptophan at this position constitutes a determinant structural or/and conformational feature of the bacterial EL4Cter. These characteristics outweigh the aromatic nature of the W355 side chain, though this may contribute to the aromatic cradle with the nearby essential Y444 in the MtPMT catalytic pocket.

Similarly, the importance of tryptophan W362, conserved in both bacterial and eukaryote PMTs, was illustrated by the absence of FascHis mannosylation in the W362A mutant (Fig. 5d). It is worth noting that, although significant, the partial reduction of about 50% of the expression of the mutated enzyme seems insufficient to explain the complete loss of activity associated with the mutation. This observation is further supported by the poor correlation between the low activity of the W362F mutated enzyme and its almost completely restored expression (Fig. 5e). Again, this demonstrated that the apolar side chain of the aromatic tryptophan W362 is probably crucial for secondary structure formation of the palindromic α-helix ($^{359}$PWTWP$^{363}$). According to the MtPMT models, this helix, planar to the enzyme periplasmic interface, may strengthen the position of the β-sheet

hairpin through anchorage to the hydrophobic core of the protein. The decisive role of this palindromic motif was further supported by a significant reduction (more than 55%) of FasC mannosylation following alanine substitution of P359 located at the N-terminal end of the alpha helix (Fig. 5d). Moreover, the requirement for stringent spatial anchoring of both extremities of the EL4Cter hairpin is fully consistent with the observed deleterious impact of mutating the conserved proline P396 located upstream TMH8 at the opposite end of the hairpin (Fig. S9). Indeed, substituting the proline with an alanine, more permissive in terms of rotational mobility, led to a greater than 95% reduction in the normalized rate of binding of ConA to FasC (Fig. 5d), underlining the importance of the torsion angle restriction of the peptide backbone imposed by the proline's rigid pyrrolidine ring. It is noteworthy that these crucial residues, P359, W362, and P396, are conserved from mycobacteria to yeast and humans, and match three characteristic amino acid components of the WW module signature.

To further investigate the functional homology of the EL4Cter with the WW domain, we mutated the highly conserved Y371 residue located in the first β-strand of EL4Cter. This tyrosine was hypothesized to correspond to the fourth component of the WW domain signature. Substituting Y371 with alanine, a mutation that disrupts the aromatic character of this residue,

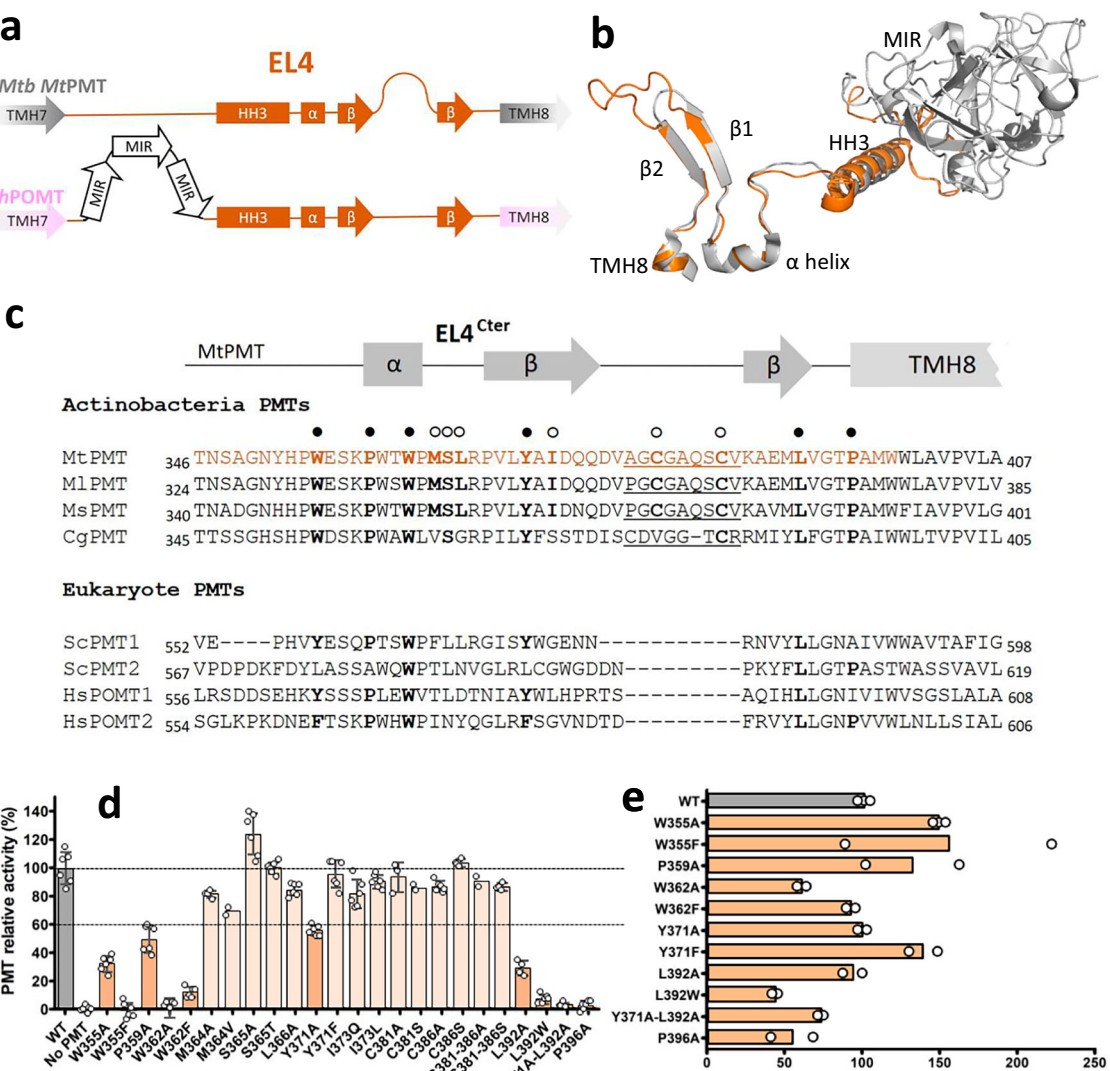

**Fig. 5 | Organization and essential function of MtPMT external loop 4 exposed at the bacterial periplasmic interface. a** Comparative organization scheme of the EL4 outer loop secondary structures of MtPMT and human hPOMTs depicting important divergences between bacterial and eukaryotic PMTs (segment size is not representative of the length of the corresponding sequences). **b** superposition of MtPMT AlphaFold 3 model (orange) and ScPMT1 (gray) EL4s showing the 3D structural similarity of the shared downstream domain of EL4s, except the eukaryotic MIR domain. **c** alignments of the MtPMT EL4 C terminal (EL4cter) sequences (in orange) with representative corresponding domains of actinobacterial PMTs and eukaryotic PMTs. The amino acids (in bold) generally conserved in PMTs or, more specifically, in mycobacterial PMTs and addressed herein are respectively annotated with full or empty circles. The insertions of a 9 supplementary amino acid found in actinobacteria PMTs are underscored (Mt: *M. tuberculosis*, Ml: *M. leprae*, Ms: *M. smegmatis*, Cg: *C. glutamicum*, Sc: *S. cerevisiae*, Hs: *H. sapiens*). **d** In situ site-directed mutational evidence that EL4cter is essential for MtPMT enzymatic activity. Effect of disruptive/rescuing mutagenesis of selected amino acids conserved in PMTs (dark orange) or specific of mycobacterial PMTs (light orange) on MtPMT relative activity. Dotted lines represent thresholds at 100% or 60% of activity, bars represent the mean, and the error bars figure the standard error of the mean (s.e.m.). MtPMT mutants with an activity less than or equal to 60% were selected for further study. **e** phosphatase activity of MtPMT-PhoA fusion proteins of MtPMT mutants with relative activity <60% supporting that the loss of activity is not due to an expression default of the mutated enzyme (PhoA activity was quantified on two to three independent clones of each strain).

resulted in a partial (40%) reduction in FasC mannosylation. Conversely, replacing Y371 with phenylalanine—a conservative substitution found at this position in other PMT homologs, including HsPOMT2—maintained mannosylation levels (Fig. 5d). These findings suggest that while Y371 is not strictly essential, the presence of an aromatic residue at this position is critical for full MtPMT activity.

Additionally, we examined the role of L392, a PMT's conserved residue located at the end of the second β-strand of MtPMT EL4cter. This leucine occupies a position analogous to the second tryptophan in most WW domains. Substituting L392 with alanine (L392A) or tryptophan (L392W) caused significant reductions in FasC mannosylation levels, by 73% and 85%, respectively (Fig. 5d). These results highlight the importance of the hydrophobic character and steric hindrance of L392 for MtPMT activity.

Interestingly, an analogous isoleucine residue in the C-terminal region of the β2-strand of Pin1's WW domain has been reported to stabilize its structure. This observation underscores the potential structural and functional parallel between the respective roles of these leucines in these motifs[72]. The functional relevance of this residue in the bacterial enzyme is also confirmed by the cumulative effect of the Y371A-L392A double mutation that results in the complete abolition of the mannosyl transferase activity of MtPMT. Thus, the deleterious impact of the L392 mutations confirmed the importance of this residue alongside P359, W362, Y371, and P396 as a determining component of WW-like domain signature. These data add to the primary sequence and β-sheet secondary structural homologies and further support the functional analogy between the EL4cter domain and WW PRM-binding domains[69].

## In-vitro analysis of the EL4 C-terminal domain with potential peptide ligands

Interestingly, WW domains are considered to be the smallest naturally occurring protein fragments capable of spontaneous folding and autonomous interaction with PRM-containing peptides when isolated in solution[70,73]. These remarkable properties prompted us to investigate whether the isolated EL4Cter peptide could spontaneously adopt a β-sheet fold similar to that predicted by the 3D MtPMT model. To address this question, we probed an EL4Cter* peptide where cysteines are replaced with alanine residues to prevent intermolecular aggregation due to disulfide bridge formation.

The far-ultraviolet circular dichroism (CD) spectrum of the synthetic peptide EL4Cter* in solution showed a single broad absorption at 195–200 nm, often attributed to the presence of short and irregular β-strands (Fig. S10)[74]. However, the absence of β-sheet characteristic negative peak at 220–230 nm does not allow to conclude on the EL4Cter* preferential folding[74]. ¹H NMR analysis also failed to confirm the WW domain's characteristic β-sheet conformation for EL4Cter* in solution. Indeed, the cluttering of the NH-CαH scalar correlation region of the EL4Cter*'s ¹H TOCSY 2D NMR spectrum indicated a weak dispersion of amide and Cα protons resonances, strongly suggesting that the peptide backbone adopts a random coil configuration[70,73] (Fig. S11).

Despite this apparent lack of ordered conformation, we tested whether the isolated EL4Cter* peptide could interact autonomously with a typical (AP[S/T])-rich ligand peptide containing a mannosylation site for the bacterial PMT. To address this, we conducted preliminary isothermal titration calorimetry (ITC) experiments to assess the ability of the EL4Cter* peptide in solution to interact with the N-terminal peptide P40–54 (Apa1, [DPEPAPPVPTTAASP]) derived from the *M. tuberculosis* Apa antigenic mannoprotein—an established acceptor ligand for MtPMT in a cell-free assay[32].

No binding was detected at temperatures below 20 °C, prompting further ITC measurements at elevated temperatures. At 37 °C, the ITC binding isotherm (Fig. S12a) obtained for the EL4Cter*–Apa1 interaction showed a positive enthalpy change (ΔH) (Fig. S12b), suggesting that hydrophobic interactions primarily drive this binding event.

To examine whether the behavior of the isolated EL4Cter* peptide in solution might reflect the consequences of an inactivating in vivo mutation, we analyzed a W10A variant of EL4Cter* (referred to as (W10A)EL4Cter*), which mimics the W362A substitution in MtPMT known to abolish enzymatic mannosyltransferase activity. The ITC binding isotherm for the (W10A)EL4Cter*–Apa1 interaction exhibited marked differences in its thermodynamic profile, characterized by an apparently exothermic binding pattern (Fig. S12c, d). These differences suggest that the W10A substitution alters the way the peptide engages its ligand, mirroring the functional impairment observed in the full-length MtPMT W362A mutant in vivo.

However, the high content of tryptophan residues in EL4Cter* significantly reduced its solubility, restricting ITC experiments to low peptide concentrations and limiting accurate measurements. This poor solubility, combined with the low apparent affinity of the interaction, constrained our capacity to further dissect the chemical determinants of binding via ITC. As an alternative approach, we investigated whether potential peptide substrates could alter the thermal denaturation profile of EL4Cter* in solution. In the absence of ligand, differential scanning fluorescence curves showed typical features of an unfolded species (relatively flat and with high initial 350/300 ratio reflecting the presence exposed Trp residues). In contrast, in the presence of the Apa1 peptide, the EL4Cter* curves showed significant changes in the heat-induced variations of the 350/330 nm fluorescence intensity ratio curves (Fig. 6a). Moreover, there is the appearance of a partial plateau with a concomitant inflection point when the Apa1 concentration is increased above 1 mM. Both the variation of the fluorescence ratio and the inflexion temperature of the curves (Fig. 6b) were clearly correlated with Apa1 ligand concentrations. The increase of the 350/330 nm fluorescence ratio resulting from the red shift of the EL4Cter* intrinsic fluorescence, reflected increased solvation of tryptophan residues as the ligand

concentration was augmented. Such modifications of the protic environment of the tryptophan indole nuclei evidenced a ligand-induced alteration of the EL4Cter*conformation that can be reasonably ascribed to an interaction with the Apa1 peptide.

To validate the ability of the isolated EL4Cter* peptide to interact with putative ligand, we probed an additional synthetic ligand (FasC) corresponding to the octadecapeptide [EPETTTEAEPTVEIPDPQ] of the major glycopeptide P30-47 of the *M. smegmatis* FasC protein that is naturally mannosylated by MtPMT[9,45]. As shown in Fig. 6c, the fluorescence ratio curves at 350/330 nm of the EL4Cter* peptide were also affected by the presence of the FasC peptide in solution. Again, such alteration, similar to that observed with the Apa1 peptide, supported an interaction between the isolated EL4Cter* peptide and the FasC P30-47 acceptor substrate. This result is consistent with the capacity of MtPMT to mannosylate the heterologous FasC protein in vivo when expressed ectopically in *M. smegmatis*.

In order to confirm the specificity and proline dependence of the above interaction, we tested a synthetic pentadecapeptide Apa2 corresponding to the internal P171–185 peptide of Mtb Apa mannoprotein. The Apa2 peptide sequence [DQKLYASAEATDSKA] is devoid of proline but contains both serine and threonine residues that have never been reported to be naturally glycosylated. Interestingly, this peptide showed no significant effect on the thermal denaturation profile of EL4Cter* (Fig. 6c). Therefore, the lack of molecular interaction between EL4Cter* and Apa2 peptide strongly suggests that the interaction between isolated EL4Cter* and ligand peptides in solution is dependent on the rigid proline pyrrolidine ring constraining the ligand peptide backbone conformation around the acceptor S/T. These results support the hypothesis of the in situ involvement of the EL4Cter hairpin in the recognition of (P[S/T])-containing target acceptor peptide for mannosylation by MtPMT.

Next, we assessed the functional impact of the W10A substitution in the synthetic EL4Cter* peptide (referred to as (W10A)EL4Cter*) on its interaction with the Apa1 ligand. Notably, the presence of Apa1 induced a clear shift in the 350/330 nm fluorescence ratio curve of the (W10A)EL4Cter* peptide, indicating that this mutant variant retains some ability to interact with Apa1 despite the W10 substitution (Fig. 6d). However, the observed fluorescence shift did not exhibit a dose-dependent response with increasing Apa1 concentrations. This lack of dose effect suggests that the W10A substitution significantly compromises the strength or specificity of the interaction. These findings, confirming the ITC observation, are consistent with the deleterious effect of the analogous W362A mutation in MtPMT, which abolishes enzymatic activity in vivo. Together, these results support the conclusion that the EL4Cter module plays a critical role in mediating interactions between MtPMT and its acceptor protein substrates in vivo.

Finally, to further challenge our hypothesis, we tested whether isolated EL4Cter* could also interact in solution with a pyrrole-containing peptide biomimetic (Z^x95) that was found to reduce FasC mannosylation in vivo[45]. Interestingly, consistent with the Z^x95 inhibitory effect of MtPMT activity in-situ, we observed a dose-dependent deviation of EL4Cter* thermal denaturation curves induced by the tested chemical (Fig. 6e). The molecular interactions between Z^x95 and the EL4Cter* peptide in solution corroborate the inhibitory effect observed on MtPMT in vivo and suggest that this compound can act as a biomimetic competitor for acceptor substrate peptide binding to the EL4-bordering pocket of the enzyme active site.

Taken together, these data strongly support our hypothesis that the MtPMT EL4Cter non-catalytic domain may assist in the recognition of proline-rich domains through a mechanism resembling that of WW-like domains.

## Discussion

Targeting bacterial MtPMT while sparing the essential human POMTs offers a potentially effective adjuvant therapeutic strategy that would support conventional anti-tuberculosis treatments by weakening pathogen fitness. However, the search for and development of MtPMT selective inhibitors to implement this approach requires a detailed understanding of

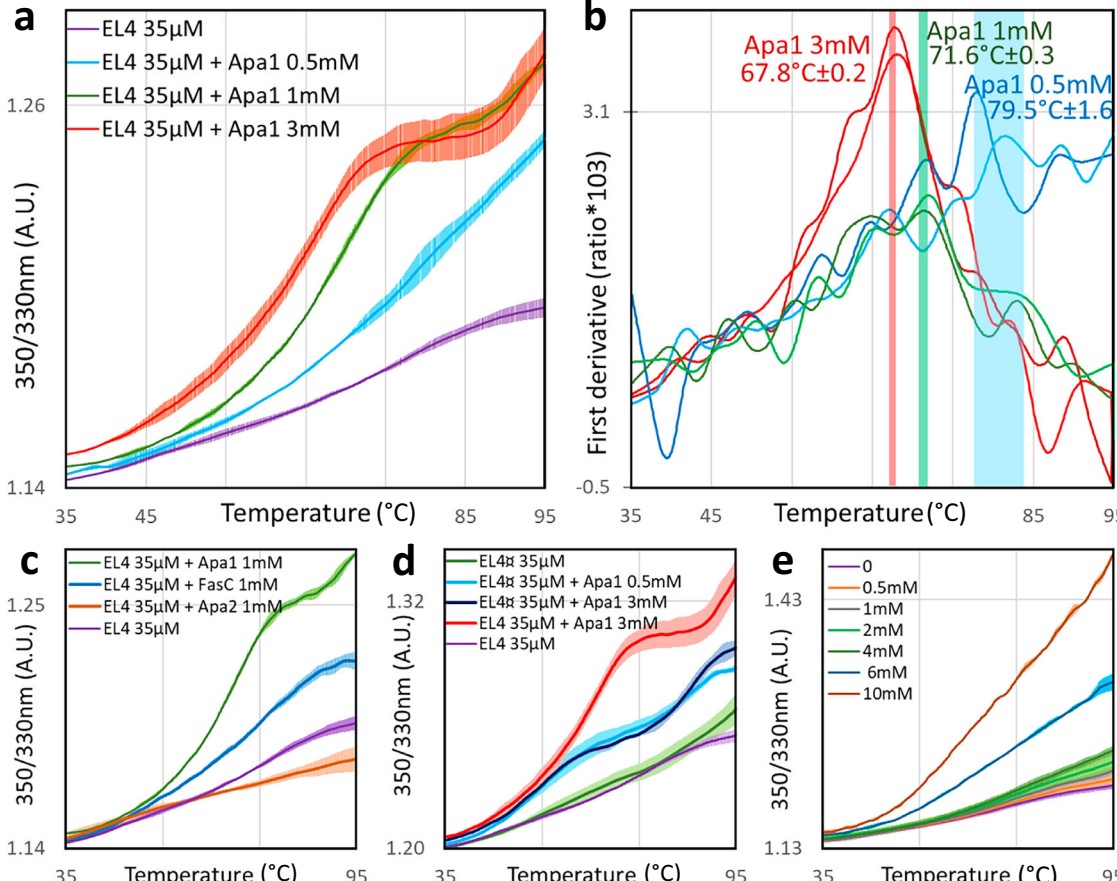

**Fig. 6 | NanoDSF thermal shift analysis showing dose-dependent specific alterations of the EL4Cter\* peptide thermal stability in presence of ligands.**
**a** Melting curves by nanoDSF of the EL4Cter\* (35 μM) in the absence (purple curve) or in the presence of increasing concentrations of Apa1 peptide (Shown is the mean of three independent measurements (plain trace) ± SDM at each point (hatched trace)). **b** First derivative of the measured fluorescence ratio of the EL4Cter\*(35 μM) in the presence of increasing concentrations of Apa1 (shown are two representative curves at each concentration), the mean ± SDM of the inflection points of the curves are indicated by vertical bars colored according the curves. **c** Melting curves of the EL4Cter\* (35 μM) in presence of 1 mM of different peptide ligands (Apa1 in green, FasC P30–43 in blue and Apa2 in red). **d** Melting curve of the EL4Cter\* Wild type (35 μM) in the presence of Apa1 (3 mM) (red curve) compared to that of the (W10A) EL4Cter\* (EL4¤) in absence (green curve) or in presence of increasing concentration of Apa1 peptide (0.5 mM: light blue; 3 mM: marine blue). **e** Melting curves of the EL4Cter\* (35 μM) in presence of increasing concentration of Z^x95. Shown is the mean of three independent measurements (plain trace) ± SDM at each point (hatched trace). For clarity, the fluorescence ratio curves presented were fitted by linear translation adjustment to make their respective origin value to coincide.

the structural and functional differences between the bacterial and human enzymes. To this end, we conducted an extensive structure-function analysis of bacterial MtPMT using a combination of molecular modeling, mutagenesis, and in cellulo and in vitro functional analyses.

By analogy to eukaryotic PMTs, previous work has assumed that bacterial MtPMT has a characteristic GT-C architecture with its amino- and carboxy-terminal ends located in opposite compartments relative to the plane of the bacterial plasma membrane. By expressing functionally competent chimeras of the enzyme fused to reporter proteins sensitive to specific subcellular compartments, we experimentally localized the N and C-termini of MtPMT to the cytosol and the periplasm, respectively. We thus confirmed that the bacterial enzyme adopts functional topology and orientation analogous to its eukaryotic counterpart. However, this topology also differs since the MtPMT C-terminus stands in the extra-cellular periplasmic space rather than the ER lumen, as observed for the eukaryotic enzymes.

According this orientation, the catalytic amino acid D74 and its associated three highly conserved amino acids D176, R441, and Y444 that constitute the reactive centers of most GT-Cs[21], were expected to localize to the periplasmic face of the enzyme. We therefore verified and demonstrated that these four residues are also essential for the activity of the bacterial enzyme. From a mechanistic point of view, the relative location of the

conserved aspartic acids D74 and D176, spatially close enough to catalyze an acid/base reaction, is consistent with an Sn2 glycosyl transfer reaction, as found in most inverting glycosidases (Fig. 3e)[22,52]. In this configuration, the catalytic aspartic acid D74 on EL1 of MtPMT would play the role of a base by facilitating the deprotonation of the hydroxyl group of the acceptor S/T residue[5,75], while the acidic residue D176, on EL2, could assist the transfer reaction as a 'proton donor' to facilitate the departure of the lipid-phosphate group. In addition, residues Y444 and R441 on EL5, which are close to the latter, would help to stabilize the S/T acceptor and the phosphate of the sugar-donating lipid, via hydrophobic and ionic interactions[29]. Therefore, the highly conserved relative spatial positioning of these 4 residues in the bacterial enzyme supports a mechanism for the mannose transfer reaction, probably identical to that proposed for eukaryotic PMTs, except that mannosylation in bacteria must occur outside the cell.

Finally, we have shown that it is possible to chemically rescue the activity of a genetically inactivated enzyme by providing cells with small molecules that can compensate for the deficiency caused by the R441A mutation. These original experiments clearly demonstrate the druggability of MtPMT in cellulo. However, given the similarity of the catalytic mechanism, it seemed difficult to envisage selectively targeting the catalytic activity of the bacterial enzyme without affecting that of the human enzyme.

On the other hand, considering that the human POMT1 and POMT2 orthologs have distinct protein targets[46,76], we assumed that MtPMT might also exhibit preferential selectivity for acceptor substrates of bacterial origin. Moreover, the fact that only a limited fraction of serine and threonine-containing sequences are naturally mannosylated suggests the existence of a mechanism for selective recognition of the acceptor substrate sequence for mannosylation. However, broad characterization of mannosylated proteins from *S. cerevisiae* failed to identify a recurring feature of mannosylated peptides that could explain why some S/Ts are mannosylated while others are not[76,77]. In contrast, similar analysis of mannosylated peptides in mycobacteria allowed us to readily evidence an over-occurrence of proline residues in the vicinity of the serines or threonines mannosylated by MtPMT[58]. This preference of MtPMT for the proline-rich peptide is, potentially, a specific and valuable feature for the selective targeting of the bacterial enzyme. Therefore, we focused on deciphering the molecular basis of selective recognition of acceptor substrates by PMTs[2,22,24].

Although a few GT-C structures have been solved with an acceptor substrate, they primarily reveal the network of bonds coordinating the reactive group of the acceptor within the catalytic center. However, these structures fall short of defining the global binding mode of the acceptor substrate, although they highlighted the second largest soluble loop, namely EL4, as a key structural element in its recognition[29,75,78–81].

In PMTs, this loop connects the prototypical PMT domain comprising TMHs 1 to 7 to the variable C-terminal module (PMT_4TMC) and invariably includes the highly conserved HH3 helix (Figs. 1d and S3). The latter forms a bridge parallel to the plane of the membrane, thus delimiting the groove through which the polar head of the sugar-donating phospho-isoprenoid substrate accesses the catalytic site. In addition, according to MtPMT conformational models, the EL4's C-terminal domain downstream of HH3 adopts a hairpin fold. This unique β-strand secondary structure faces the domain of EL1 upstream of HH1, forming a channel leading to the active site harboring the catalytic amino acid D74. Interestingly, the cryoEM structure of the ScPMT1-ScPMT2 heterocomplex revealed the presence of the tripeptide PYT, an analogue of the substrate acceptor, in the homologous groove of ScPMT1.

Given the above converging topological evidence, we hypothesized that MtPMT EL4Cter could contribute to the specific interaction with target peptide substrates. Deeper inspection of the MtPMT putative interface with the acceptor substrate revealed substantial similarities with proline-rich motif binding domains, notably the presence of an aromatic cradle together with a motif closely resembling the 5-amino acid signature of WW protein domains in the EL4Cter portion. WW domains are non-catalytic modules of approximately 40–50 amino acids defined by the presence of 2 highly conserved tryptophans bordered by conserved prolines flanking a strictly conserved inner tyrosine or phenylalanine aromatic residue. These domains, characterized by their 3-stranded antiparallel β-sheet conformation[61,68,73,82,83], are considered major protein-protein interaction modules that have been classified into 4 groups based on the proline-rich consensus sequences they recognize. Notably, WW domains of group IV specifically recognize peptides containing phosphorylated and un-phosphorylated S/T-P sequences similar to those found in mycobacterial mannosylated peptides[61,84]. WW domains have been described in a great variety of intracellular proteins, ranging from cytoskeletal to signal-transduction and regulatory proteins. So far, these have been exclusively characterized in eukaryotic proteins, although the WW primary sequence signature is detected by automatic annotation algorithms in various bacterial genomic coding sequences, particularly in bacteria of the order Mycobacteriales (Supplementary data 2). Closer inspection shows that in these organisms, automatic annotations almost exclusively point to proteins of the ferredoxin-dependent glutamate synthase (Gltb) family, including the Mtb *rv3859c* gene product (inferred from homology). However, examination of the AlphaFold model (AF-P96218-F1) of Mtb Gltb shows that the WW signature segment likely adopts in these complex proteins, an α-helical fold, invalidating any functional approximation with eukaryotic WW domains (Fig. 7).

On the contrary, here we found that the EL4Cter domain of MtPMT possesses most of the structural and functional properties characteristic of WW domains, albeit with some imperfections. In fact, we first demonstrated that the 4 amino acids that evoke the WW domain signature in the EL4Cter sequence are crucial for the activity of MtPMT *in cellulo*. This result not only demonstrates the essential nature of the EL4Cter domain for MtPMT activity but also supports a WW-like selective recognition function for this motif. Second, this view is further supported by the similarity of the β-hairpin secondary structure of the EL4Cter domain and the prototypical three β-stranded conformation of WW domains. Third, we were able to show that the EL4Cter domain shares the perhaps more remarkable feature of WW domains, namely their ability to interact with their ligands when isolated in solution. Indeed, in vitro experiments clearly demonstrated that an analog of the EL4Cter peptide readily interacts with natural target peptides in solution. Furthermore, these experiments confirmed that the interaction was conditioned by the presence of proline in the sequence of the target peptide, in full agreement with the preferential composition of myco-bacterial peptides mannosylated by MtPMT in vivo. There are also notable differences between the EL4Cter domain and the WW modules, in particular the incompleteness of the signature and the absence of a third β-strand. However, these features have also been reported for some non-canonical WW domains[68,85,86]. Thus, our functional and molecular data strongly support the essential role of the MtPMT EL4Cter domain in the selection of proline-containing target peptides according to a mechanism that is comparable to that of eukaryotic WW modules. Interestingly, from a phylogenetic perspective, alignment of the primary sequences of the referenced eukaryotic WW domains and EL4Cter domains of different mycobacterial PMTs suggests that these motifs may have evolved independently from a common precursor while retaining similar functionalities (Fig. 7 and Supplementary data 2).

From a practical point of view, the presence of similar WW motifs in human or yeast PMT homologs may significantly hamper the research and development of selective inhibitors of bacterial MtPMT. However, our analyses suggest a major difference regarding the apparent preference of MtPMT for proline-rich target sequences, which is not observed in eukaryotic PMTs[2,77,87]. Several hypotheses can be proposed to explain this difference in target peptide selectivity between eukaryotic and prokaryotic PMTs. The first is the likely cooperative contribution of other protein motifs adjacent to the catalytic site, such as the proline cradle, the EL1 loop, or the MIR domains of eukaryotic PMTs, in addition to the EL4Cter site[2,20]. Alternatively, the presence of competing PMTs with different acceptor specificities in eukaryotes has been proposed to explain the apparent lack of selectivity for proline-rich peptides in eukaryotes[76,77]. Regardless, this peculiarity of bacterial PMTs provides a potential gateway to selective targeting strategies for MtPMT.

In conclusion, our analysis sheds new light on the possible mechanisms underlying the selection and binding of the acceptor peptide substrate for mannosylation by bacterial MtPMT. In addition to providing evidence for the presence of a eukaryotic WW-like domain in bacteria, this work also proves the possibility of drugging bacterial MtPMT, confirming MtPMT as a valuable drug target with high therapeutic potential. However, several fundamental questions remain to be addressed, including (1) whether MtPMT dimerizes like its eukaryotic counterparts, (2) how bacterial PMTs might compensate for the absence of MIR domains, (3) whether MtPMT physically associates with the Sec complex as a formal component of the translocon machinery, and finally (4) how the acceptor sequence of the target protein is translocated, reaches the periplasm, and arrives at the reaction site of membrane-embedded MtPMT[25]. A deeper understanding of these complex mechanisms will further pave the way for a rational search for MtPMT-selective inhibitors.

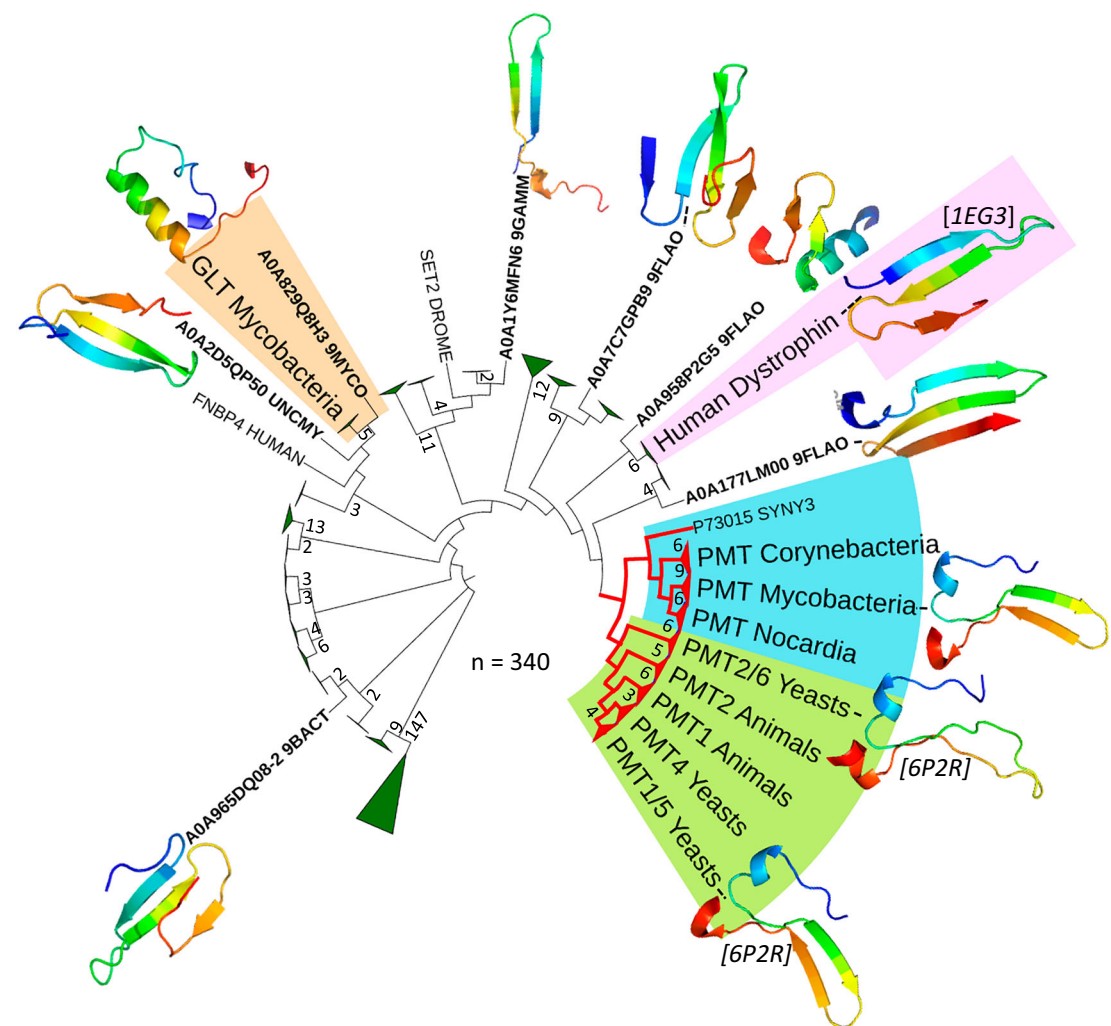

**Fig. 7 | Phylogenetic tree illustrating the potential evolutionary relationship between PRM-recognition WW domains (black branches) and the C-terminal domains of EL4 PMT loops (red branches).** The Muscle[5] multiple sequence alignment analysis suggests that PMTs and the human dystrophin clade may share a common ancestor. For clarity, clades with an average branch length distance to their leaves below 0.35 have been collapsed, except for the PMT clade, for which leaves are grouped based on either "Mycobacteriales" genus or the respective homology of eukaryotic PMT orthologs. (Collapsed clades are represented by triangles annotated with the number of corresponding leaves; figure generated with iTOLv7). The AlphaFold-predicted conformations of WW domains from representative putative prokaryotic proteins are shown alongside the canonical WW motif of human dystrophin and the EL4Cter domains of MtPMT and ScPMTs (italicized annotations in brackets correspond to PDB accession codes).

## Materials and methods
### Enzymes and reagents
Phusion polymerase (F530S), restriction enzymes, T4 DNA ligase (EL0011), and low melting-point agarose (16520050) were from Thermo Fisher Scientifics. Electro-competent *E. coli* TOP10 strains (C404003), Super Fi DNA polymerase (12361010), and anti-His C-term-HRP antibody (MA180218) were purchased from Invitrogen. NEBuilder HiFi DNA Assembly Cloning Kit (E2621S) and *E. coli* Stellar chemically competent cells (636763) were purchased, respectively, from New England Biolabs and Takara Bio. Luria Broth (L3022), M9 minimal medium (M6030), and PKH67 (PKH67GL) were from Sigma-Aldrich. Ni-NTA spin columns (31314) and plasmid miniprep kits (27106) were obtained from Qiagen. 96-well microplates coated with anti-His antibody (L00440C) were acquired from Genscript. TMB solution (00-4201-56) was purchased from eBioscience. Z$^{x95}$ (Z66495095 (no CAS, IUPAC (2E)-3-[2,5-dimethyl-1-(5-methyl-1,2-oxazol-3-yl)-1H-pyrrol-3-yl]-1-[2-(thiophen-2-yl)pyrrolidin-1-yl]prop-2-en-1-one) came from the Enamine fragment library. Oligonucleotides were synthesized by Eurofins Genomics and are detailed in Table S2; this company also provided sequencing services. Synthetic EL4Cter* peptides (≥92% purity) and

putative ligands (≥96% purity) were respectively obtained from sb-PEPTIDE (SmartBioscience SAS) and ProteoGenix SAS.

### DNA constructs
Targeted mutations in the MtPMT (Rv1002c) gene were generated through directed mutagenesis using complementary primers designed to introduce the desired mutations alongside silent mutations removing or adding restriction sites (refer to Table S2 for primer sequences). Mutations were incorporated into the MtPMT gene in the pMW158 plasmid through whole plasmid PCR with Phusion polymerase, using a high-GC buffer with 3% DMSO. PCR products were digested with DpnI (ThermoFisher) and then used to transform electro-competent *E. coli* TOP10 strains (Invitrogen).

Fusion proteins were constructed from linear DNA fragments amplified by PCR with Platinum SuperFi II DNA Polymerase (Invitrogen) and primers listed in Table S2 from plasmids listed in Table S3. Linear amplicons were merged with the NEBuilder HiFi DNA Assembly Cloning Kit (New England Biolabs) and directly transformed into *E. coli* Stellar strains (Takara Bio). Constructed plasmids were amplified and purified from overnight *E. coli* cultures. Construct validation included restriction profile analysis and subsequent sequencing of the gene.

The control plasmids employed for cytoplasmic and periplasmic expression of the PhoA enzyme were constructed by inserting KatG or MmpS4 genes, along with PhoA, between the NdeI and HindIII restriction sites of the pWM158 vector using T4 DNA ligase.

Notably, the MtPMT sequence varies across different databases, with the NCBI Ref. Seq. NP_215518.1 featuring a shortened N-terminal end of 19 amino acids compared to the UniProt P9WN05 sequence. In this study, we used the shortest version of the enzyme, as the absence of extended N-terminal end does not affect the enzymatic activity (45).

### *Mycobacterium Smegmatis* transformation and culture

Plasmids used are listed in Tables S3. Competent *M. smegmatis* strains were prepared as follows: Bacteria were cultured for a week in 10 mL of LB medium with Tween 80 0.05% (called LBT medium) until reaching an OD600nm of 1. Afterward, 500 μL of the culture was used to inoculate 50 mL of LBT medium and cultured to achieve an OD600 nm of 0.2–0.8. The cells were then washed three times with decreasing volumes (50 mL, 15 mL, and 1 mL) of cold Washing Buffer (Tween 80 0.05%, Glycerol 10%), with centrifugation (8 min, 4 °C, $500 \times g$) between each wash. Washed competent cells were divided into 200 μl aliquots. Each aliquot was mixed with 1 μL of plasmid (200 ng/μL) and subjected to electroporation (2500 V, 25 μf, 200 Ω). Following transformation, bacteria were incubated in LBT medium for 3 h at 37 °C with shaking (180 rpm). Transformed bacteria were plated on LB-agar with appropriate antibiotics and incubated at 37 °C for 5 days. Isolated colonies were cultured in LBT medium with antibiotics at 37 °C for 9 days, until OD600nm reached 1. For further analysis, cells were centrifuged (10 min, 4 °C, $500 \times g$) and the supernatants and pellets stored separately at −20 °C.

### Structured illumination microscopy of live bacteria

Mycobacterial membranes were labeled with a green fluorescent dye, PKH67 (Sigma-Aldrich). A mycobacterial culture was pelleted and washed once with PBS before resuspending the pellet in 100 μL of PKH solution (1 μL of fluorophore and 99 μL of diluent C). Staining was carried out in the dark at room temperature for 15 min with periodic mixing. To stop the reaction, 200 μL of BSA was added. The bacteria were then centrifuged, and the pellet was resuspended in M9 minimal medium (Sigma). The cell suspension was applied to an agarose pad consisting of 1% (w/v) low-melting-point agarose (Fisher Scientific) in M9 minimal medium. Images were acquired using the Zeiss ELYRA 7 Sim Lattice microscope equipped with a Zeiss 63 × NA = 1.4 objective. Image analysis was performed using Zeiss's ZEN software.

### Alkaline phosphatase activity assessment

PhoA activity in *M. smegmatis* strains was measured after liquid culture (LB with 0.05% Tween 80) incubated at 37 °C, 180 rpm for 5 days. To prevent spontaneous folding and activation of cytoplasmic PhoA, 10 mM iodoacetamide was introduced 20 min prior to harvesting and maintained throughout the experiment. After centrifugation (5000 rpm, 10 min, 4 °C) and two washes with reaction buffer (1 M Tris pH 8, 0.05% Tween 80, 10 mM iodoacetamide, 10 mM MgCl₂), cells were normalized to 0.4 g/mL. 200 μL of the standardized cell suspensions were mixed with 1 mL of reaction buffer supplemented with 10 mM of para-nitrophenylphosphate and incubated at 37 °C in the dark, with gentle agitation. A two-hour kinetic assay was conducted, followed by the addition of 50 μL of stop solution (1 M K₂HPO₄) to each cell suspension sample. To remove cells and debris, samples were centrifuged (13,000 rpm, 10 min, at 4 °C), and the OD420nm of the resulting supernatants was measured to monitor para-nitrophenylphosphate hydrolysis by PhoA. Additionally, alkaline phosphatase activity was phenotypically detected on solid LB-agar culture supplemented with 60 μg/mL BCIP (5-brom-4-chloro-3'-indolyphosphate p-toluidine salt) and appropriate antibiotics. Blue colonies appeared after 7 days.

### Enzyme-linked lectin assay (ELLA)

Glycosylation of FasC^His in the culture supernatant was estimated by ELLA as described previously[45]. First, to assess saturable amount of FasC^His for

ELLA mannosylation relative quantification, FasC^His concentration from culture supernatants has been quantified by direct-ELISA. For this, wells of a 96 well-plate were filled with 50 μL of culture supernatant, diluted 100× in PBS, and mixed with 50 μL of carbonate coating buffer (Na2CO3 0.53%/NaHCO3 0.42%, pH 9.6) and then incubated overnight at room temperature. After washing 3 times with PBS, the plates were saturated for 1 h at room temperature with 200 μL/well PBS containing 0.05% Tween 20 and 5% milk. After 3 washes with PBS containing 0.05% Tween 20, the plate was incubated for 2 h at room temperature with 50 μL/well of anti-His antibody (anti-His C-term-HRP antibody, Invitrogen), diluted 1000× in PBS containing 0.05% Tween 20 and 0.5% milk. Four washes with PBS were performed before revelation. Subsequently, supernatants with normalized FasC^His concentrations were analyzed for mannosylation by ELLA using 96-well microplates coated with anti-His antibody (Genscript, ref L00440c) were used. Plates filled with 100 μL/well of culture supernatant (v/v in PBS) were incubated for 3 h at room temperature and then washed 4 times with PBS containing 2% Tween 20. 100 μL of Concanavalin A-HRP solution (Sigma Aldrich, L6397) (0.5 μg/mL ConA-HRP, 1 mM CaCl2, 1 mM MgCl2, 1 mM MnCl2, 0.05% Tween20, in PBS) was added and the plate was incubated at room temperature for 2 h before 4 washes with PBS. Relative values of 100% and 0% were attributed to the amount of ConA binding observed for the WT and No PMT *M. smegmatis* control strains, respectively. For both assays, revelation was performed with 50 μL/well of TMB solution (eBioscience). 25 μL/well of stop solution (1 mM sulfuric acid) was added after 20 min incubation. Absorbances were measured at 450 nm using the BMG LABTECH Clariostar microplate reader.

### LC-ESI-MS analysis of FasC^His glycoforms

Analysis of purified FasC^His glycoforms was conducted using entire protein mass spectrometry after purification from culture supernatant using Ni-NTA spin columns as described previously in ref. 45. NanoLC-MS analyses were performed by injecting 5 μL of purified proteins diluted in 2% acetonitrile, 0.1% trifluoro acetic acid loading buffer (final concentration ~0.4 μM), in a nanoRS UHPLC system (Dionex) equipped with a reverse-phase C4 pre-column (300 μm i.d. × 5 mm; Thermo Fisher Scientific). After 5 min of desalting at 20 μL/min in 2% ACN and 0.05% TFA, the precolumn was switched online to a home-made C4 analytical nanocolumn (75 μm i.d. × 15 cm) packed with C4 Reprosil (Cluzeau CIL). Proteins were eluted using a 38 min linear gradient from 5% to 100% B (0.2% FA in ACN) in solvent A (0.2% formic acid (FA), at a flow rate of 300 nL/min. MS scans were acquired with an LTQ-Orbitrap Velos mass spectrometer (Thermo Fisher Scientific) in positive mode in the 800–2,000 $m/z$ range with a resolution set at 60,000. Raw data of FasC^His, eluting around 25 min, were deconvoluted with the UniChrom2 module of UniDec[88] with the following parameters: m/z range: 1000–2000 Th; background subtraction: 1; bin every 1 Th; charge range: 10–25; mass range: 22,000–28,000 Da; sample mass: every 1 Da; peak FWHM: 1 Th; picking range: 100 Da, and peak detection threshold: 0.06.

### In vivo chemical rescue

Chemical rescue to restore R441A MtPMT activity was adapted from protocols previously described for culture in a 96-well microplate of such strains and treatment with putative MtPMT inhibitors[45]. 9 mL of one-week cultures of recombinant *M. smegmatis* strains carrying R441A mutations, along with control strains, were centrifuged at 4000 rpm and 4 °C for 10 min. Pellets were resuspended in 20 mL of LBT using sterile glass beads (4 mm), then centrifuged again at 800 rpm for 10 min to retain only supernatants. The OD600nm of these supernatants of dissociated cells was measured and used to initiate micro-cultures in 96-well plates. Wells containing LBT medium were inoculated at an OD600nm of 0.01. Micro-culture plates were then incubated at 37 °C with gentle daily shaking. After 48 h, OD600nm measurements were taken, and chemical compounds were added (imidazole or guanidine hydrochloride at concentrations of 2.5 mM or 5 mM). Control cultures

were grown without any treatment. FasC$^{His}$ glycosylation was then measured by ELLA on collected culture supernatant as described above.

## NMR
The EL4Cter* peptide samples were solubilized at 3.2 mg/mL in 20 mM phosphate buffer in 90/10 $H_2O/D_2O$, pH 6.5. 1D $^1H$ and 2D $^1H$-$^1H$ TOCSY (80 ms mixing time) NMR spectra were acquired on a Bruker 600 MHz spectrometer equipped with a cryoprobe at either 295 or 300 K.

## Circular dichroism
The secondary structure of the EL4Cter* peptide was studied using the Jasco circular dichroism spectrometer. The peptide was dissolved in water, and measurements were taken at 25 °C. The results are expressed as the average residue ellipticity [Θ] (in deg.cm^2/dmol) as a function of wavelength.

## Isothermal titration calorimetry (ITC)
Peptide concentrations were systematically determined using a NanoDrop spectrophotometer (Thermo Scientific). For the EL4Cter* peptide, absorbance at 280 nm was measured using the protein quantification method, with an extinction coefficient ($\varepsilon$) of 24,980 $M^{-1} cm^{-1}$. For the Apa1 peptide (molecular weight: 1445 g/mol), concentrations were estimated using the A205 application. ITC experiments were conducted at 37 °C using a MicroCal ITC200 instrument (Malvern Panalytical, Malvern, UK) to monitor the interaction between EL4Cter* (15–35 μM) or the mutated W10AEL4Cter* peptide (25 μM) with the Apa1 P40–54 peptide (100–700 μM). Titrations were performed in 20 mM sodium phosphate buffer containing 50 mM NaCl and 1 mM EDTA, pH 7.4, at 37 °C. Peptide stock solutions were prepared by resuspending lyophilized synthetic peptides (Smart-Biosciences) in ultrapure water, followed by pH adjustment to 7.0. Solutions were then freeze-dried and reconstituted in ITC buffer to the required final concentrations. The titration protocol involved an initial injection of 0.4 μL of the Apa1 P40–54 peptide solution into the thermostatic sample cell containing EL4Cter* or the mutated W10AEL4Cter*, followed by 19 consecutive injections of 2 μL each (initial delay: 60 s; injection duration: 2 s; spacing between injections: 150 s). Control dilution experiments were performed by titrating the peptide ligand into buffer alone to account for the heat of dilution.

## Thermal denaturation experiments
The thermal denaturation of the EL4Cter* peptide was monitored by differential scanning fluorimetry (nanoDSF) using the 350 nm/330 nm ratio on a Tycho NT.6 Nanotemper spectrofluorometer. The EL4Cter* wild-type peptide was dissolved in MiliQ $H_2O$. Melting curves of the EL4Cter* peptide (35 μM) in the presence of either increasing concentrations of Apa1 P40–54 peptide (0.5, 1, and 3 mM), or in the presence of 1 mM of Msmeg FasC P30–47 peptide or Apa2 P171–185 peptide were recorded. In addition, the melting curves of the mutated (W10A)EL4Cter* peptide (35 μM) in the presence or absence of increasing concentrations of Apa1 P40–54 peptide (0.5 and 3 mM) were recorded.

## Phylogenetic analysis
The phylogenetic analysis was conducted on a set of non-redundant primary sequences of 294 WW domains of eukaryotic proteins (reviewed in UniprotKB; $n = 266$) plus arbitrarily selected putative bacterial proteins ($n = 28$) with a hit to the InterPro PS50020 and of 47 EL4Cter domains of eukaryotic ($n = 24$) and bacterial ($n = 23$) PMTs (Supplementary data 2). Multiple sequence alignments were realized using ClustalOmega from the "EMBL-EBI Job Dispatcher" sequence analysis tools, and the obtained Phylogenetic Tree was further pictured via iTOLv7.

## Statistics and reproducibility
Otherwise stated, bar charts and XY dot graphs data correspond to the relative enzyme activity quantified from independent biological replicates (independent bacterial cultures of unique clones). Error bars in all graphs for which $n \geq 3$ represent the standard error of the mean (s.e.m.).

In Fig. 3a inset graph, each bacterial mutant ("No Pmt", "D74A", "D74E", "D176A", and "D176E") strain's PMT activities were measured using the ELLA assay. Each condition was tested using three independent biological replicates, each comprising three technical replicates. Technical replicates were averaged within each biological replicate, resulting in three independent values per mutant strain.

To assess whether the mean PMT activity differed significantly between strains, we performed pairwise comparisons using unpaired Student's t-tests with Welch's correction to account for unequal variances. One-tailed tests were used to test specific directional hypotheses based on prior biological expectations. Pertinent differences considered statistically significant (at $p < 0.05$) are presented in Fig. 3A. No correction for multiple comparisons was applied in the main analysis, although Bonferroni-adjusted thresholds were considered for interpretative purposes. All statistical analyses were conducted using Python (SciPy library, v1.11.2).

## Reporting summary
Further information on research design is available in the Nature Portfolio Reporting Summary linked to this article.

## Data availability
All data underlying bar charts and XY dot graphs data in Figs. 2c; 3a, b, d, and 5d, e are listed in Supplementary data file 1. Peptide sequences used for Figs. 4a and 7 are available in supplementary data files 2 and 4, and numerical data for Fig. 6 and Supplementary Fig. 12 are reported in supplementary data files 3, 5 and 6, respectively. The mass spectrometry data for Fig. 2d have been deposited to the ProteomeXchange Consortium via the PRIDE[89] partner repository with the dataset identifier PXD065654".

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

## Acknowledgements

We express our gratitude to Luiz Pedro de Carvalho (UF Scripps Institute, Florida, USA) for helpful discussion, Claude Gutierrez (IPBS, Toulouse, France) for providing the mCherry plasmid, and to Christian Chalut (IPBS, Toulouse, France) for supplying the PhoA plasmids. We acknowledge the GENOTOUL-TRI-Microscopy platform for their technical support, as well as Elodie Vega (IPBS, Toulouse, France) for her expert guidance on structured illumination microscopy. We thank the PICT-ICEO facility of the Toulouse Biotechnology Institute, which is part of the Integrated Screening Platform of Toulouse (PICT, IBiSA), for providing access to circular dichroism, nanoDSF, and NMR equipment. The ITC experiments were performed on the PIM Platform of I2BC (Platform for measurements of Interactions of Macromolecules). We also acknowledge Wladimir Malaga (IPBS, Toulouse, France), David Rengel (IPBS, Toulouse, France), Virginie Nahoum (GENO-TOUL-PICT platform, IPBS, Toulouse, France), and Guy Lippens (TBI, Toulouse, France) for enriching discussions. This work was supported by the Fondation de la Recherche Médicale (Equipes FRM DEQ20180339208 to NJ.), the Fondation MSDAVENIR (grant FIGHT-TB to Nigou J.), the Fondation Rolland Garrigou pour la culture et la santé (FONROGA, for NG Master internship grant), and the Agence Nationale de la Recherche for the MUS-TART project (grant ANR-20-PAMR-0005 to NJ.). This work was also funded in part by grants from the Région Occitanie, European funds (FEDER, Fonds Européens de Développement Régional), Toulouse Métropole, and the French Ministry of Research with the Investissement d'Avenir Infrastructures Nationales en Biologie et Santé program (ProFI, Proteomics French Infrastructure project, ANR-10-INBS-08).

## Author contributions

All authors contributed extensively to the work presented in this paper. N.G. and C.R. contributed equally to the genetic and functional analyses. N.G. conducted the imaging experiments, C.R. and G.C. performed the nano-DSF analyses. V.G. performed ITC, C.Fa. realized the chemical rescue experiments, and M.G. recorded the NMR spectra. J.M. and C.Fr. realized the MS analyses. J.N. gave technical support and conceptual advice while. E.F. and M.R. designed the project, administered the experiment, and wrote the manuscript.

## Competing interests

I declare the authors have no competing interests as defined by Nature Research or other interests that might be perceived to influence the interpretation of the article.
