## [Transparent Peer Review file · Communications Biology]

Structural Insights into Mycobacterial Protein-Mannosyltransferase reveal a WW-Domain-Like Protein Motif in Bacteria.

Corresponding Author: Dr Michel Riviere

Version 0:

Reviewer comments:

Reviewer #1

(Remarks to the Author)

In the submitted manuscript, the authors have comprehensively evaluated the function of conserved residues in the mycobacterial protein mannosyl transferase. The work is very well done and is an excellent example of how papers using predicted structures should work. They generated some models and then actively grounded them in wet-lab experiments. I only have a few minor suggestions for improvements to the paper.

Since the release of AlphaFold 3, it has become possible to model peptide ligands. I would think a P8-S/T-P8 ligand, or a EPETTTEAAPTVEIPDPQ peptide co-fold on alphafoldserver.com might be interesting. I would be careful of the latter peptide, as the acidic residues may occupy the position one would expect a phosphate to sit (a brief attempt at this on my own suggests this is the case when compared to the ligand in 6P2R). Nonetheless, this may help refine some of the hypotheses (or further support them) for the function of the conserved residues. This could then be compared nicely to the model in figures 3c,4b.

Line 273: While I think this experiment is really interesting, and shows that these compounds can rescue the mutation, your data does not actually show that they “were able to diffuse through the bacterial envelope, [and] reach the mutated enzyme at the membrane.” You haven’t shown active engagement of these compounds with the protein directly. While less likely, some other series of events may be leading to this complementation. I would temper this statement to something closer to ‘Interestingly, we found that either guanidine chloride or imidazole were able to restore FasC mannosylation to approximately....’

Line 387 – I am not sure P396 is the residue you intend to discuss here? It appears to be nowhere near the other two in the predicted structure.

Minor comments:

Line 193 – analysis, not analyzes

Line 200 – ‘key’ seems redundant with ‘selected’ here, I’d use one or the other

Line 219 – in order to -> ‘in order’ can be deleted and lose no meaning while improving readability

Line 259 – odd line break

Line 264 – I wouldn’t quite call lysine and arginine ‘chemically equivalent’ I would just delete that part of the sentence

Line 377 – ‘functional folding’ is awkward, not totally sure what that means

Figure 3a – the grey text (clone b exp2) in the inset is hard to read at this size. When typeset for publication I think it will be unreadable.

Figure 3c and elsewhere – It is far easier to understand the context of residues/ligands if they are coloured by heteroatom with the residue carbons a sufficiently contrasting colour. I would not colour the lipid carbons orange as you have, as this is typically used in phosphates.

Supplemental: There should be a table with quality metrics for the predicted structures where available. With AF models it is simple to colour by pLDDT or report the average pLDDT. Admittedly the standards of reporting on models is only starting to grow, but the readers should have some sense of how much confidence the modelling software itself has in the prediction.

Reviewer #2

(Remarks to the Author)

The discovery that protein O-mannosylation (POM) also occurs in bacteria, particularly in clinically significant mycobacteria, highlights its role as a potential virulence factor (see key reference 9 in the current manuscript). This finding has spurred interest in the characterization of the corresponding enzyme, MtPMT, as a potential drug target. The appeal of MtPMT is strengthened by its structural distinction from eukaryotic PMTs, which are heterodimeric complexes. In contrast, MtPMT, encoded by a single gene, is likely to function as a monomeric protomer.

Against this background, the present study aims to elucidate MtPMT's membrane topology and structure-function relationships while evaluating its viability as a drug target. Using a comprehensive suite of bioinformatics tools, the authors predict MtPMT's membrane topology and functional sites, validating some theoretical predictions through transfection experiments in a PMT-depleted *M. smegmatis* strain.

The membrane topology experiments, designed with a well-structured set of fusion constructs, are solid and convincing and so are cellular experiments controlling for functional expression and expression levels. However, investigations into the predicted mechanistic elements and target selection features (WW-motif based recognition of Ser/Thr residue in proline-rich protein segments) lack unequivocal evidence. Additionally, some aspects of these experiments suffer from insufficient controls.

Before addressing specific weaknesses, I summarize this study as interesting but, in its current form, not meeting the standards for a high-impact publication. The proposed structure-function relationships and target selectivity studies remain largely speculative. More critically, the study does not address the key questions that arise throughout the text — questions the authors themselves succinctly outline in the final paragraph of the discussion (page 26, line 610 ff). Here a copy of the relevant passage: "(1) whether MtPMT dimerizes like its eukaryotic counterparts, (2) how bacterial PMTs might compensate for the absence of MIR domains, (3) whether MtPMT physically associates with the Sec complex as a formal component of the translocon machinery, and finally (4) how the acceptor sequence of the target protein is translocated, reaches the periplasm, and arrives at the reaction site of 618 membrane-embedded MtPMT." Significant information to these questions is essential before MtPMT can be seriously considered as a drug target.

Major points:

The authors claim to have demonstrated the "druggability" of MtPMT by compensating for the loss of the guanidinium side chain of R441 in a mutant using a guanidinium salt buffer. While I agree that this in cellulo experiment (showing 25% reconstitution) suggests accessibility of the active site pocket, the study lacks experiments that probe the functional consequences of interfering with the WW-motif, claimed to be essential for MtPMT activity. Although this element may distinguish MtPMT from host PMTs, the WW-motif is abundant in the host, and proline-rich motifs are common in mammalian proteins. These motifs play critical roles in protein-protein interactions, influencing signaling pathways, structural stabilization, and the regulation of O-glycosylation and phosphorylation. All together massively challenging the evaluation of MtPMT as a drug target.

Furthermore, in the mutational experiments conducted on the C-terminus of EL4 to probe interactions with potential ligands, it would be preferable to mutate not only one interaction partner but also confirm the findings by introducing mutations in the second partner. Taken together, the experiments described on page 18 ff. are interesting but require proper controls. The authors appear to be aware of these issues, as much of the text is framed in a tentative manner.

Minor

Page 3, line 63: It seems the cited literature is not adequate. Instead of 9,18 the citations 19,20 seem to be more fitting. In general the allocation of literature should be carefully controlled.

Mix-up of numbering in Fig. S4

Fig. 6: subnumbering is missing

ER does not need explanation but there are other truncates where the reader would profit from explanation.

Minor grammar and spelling errors throughout the text

Reviewer #3

(Remarks to the Author)

Geraud and colleagues perform a thorough structure-guided study to confirm the predicted topology of MtPMT and to functionally validate key residues for *M. tuberculosis* PMT activity. Although they do not solve the protein structure, which, due to its nature, is an extremely difficult endeavour, they use high-quality models and structural comparisons with orthologues and homologues to infer the structure of this protein and identify residues to explore. The authors performed an impressive amount of work to address these questions, including mutagenesis on active site residues and residues involved in peptide substrate recognition, which is the highlight of this work.

The identification of a WW-like domain is novel and compelling, although the absence of structural validation is a limitation. My only concern is the exclusive reliance on DSF as the biophysical validation method. This aspect would benefit from additional validation, and it is unclear whether the authors tested other biophysical methods. This limitation may reflect the low affinity of these interactions, making them challenging to measure. However, the lack of further biophysical validation is

not fundamental for the publication of this work if it cannot be obtained, but attempts should be made to explore alternative methods.

The manuscript is very well-written throughout, and references are appropriate. Figure 4b would benefit from a closer view of the EL4 and EL1 regions and higher resolution. Additionally, Line 193 "HR-MS analyzes" should be corrected to "HR-MS analysis."

Apart from these minor points, I have no further suggestions or criticisms of what is a thoughtful and well-executed work.

Version 1:

Reviewer comments:

Reviewer #1

(Remarks to the Author)

The authors have addressed my concerns.

Reviewer #2

(Remarks to the Author)

After a thorough review of the revised manuscript, and in due recognition of the fact that this referee is clearly the most distant from the presented study, which is largely based on computer-aided protein analysis, I recommend the revised manuscript for publication. Readability and comprehensibility have indeed significantly improved, even for readers who are less familiar with the topic.

Reviewer #3

(Remarks to the Author)

I am satisfied with the responses given by the authors and consider the manuscript ready for publication.

Response to the reviewers' comments (in blue)

Reviewer #1 (Remarks to the Author):

In the submitted manuscript, the authors have comprehensively evaluated the function of conserved residues in the mycobacterial protein mannosyl transferase. The work is very well done and is an excellent example of how papers using predicted structures should work. They generated some models and then actively grounded them in wet-lab experiments. I only have a few minor suggestions for improvements to the paper.

Since the release of AlphaFold 3, it has become possible to model peptide ligands. I would think a P8-S/T-P8 ligand, or a EPETTEAEPTVEIPDPQ peptide co-fold on alphafoldserver.com might be interesting. I would be careful of the latter peptide, as the acidic residues may occupy the position one would expect a phosphate to sit (a brief attempt at this on my own suggests this is the case when compared to the ligand in 6P2R). Nonetheless, this may help refine some of the hypotheses (or further support them) for the function of the conserved residues. This could then be compared nicely to the model in figures 3c,4b.

We fully agree with the reviewer's remark and have followed the suggestion by modeling the complex of the MtPMT protein (NCBI Ref. Seq. NP_215518.1) with the PYTV peptide (as described in Bai et al.) using AlphaFold 3 (ipTM = 0.61, pTM = 0.92). Figure 3c has been updated to show the superposition of the modeled PYTV peptide within the MtPMT active site and the PYT peptide as observed in the active site of ScPMT2 in the cryo-EM structure reported by Bai et al. The structural alignment (PyMOL RMSD^{MtPMT-PYTV / ScPMT2-PYT} = 2.39 Å) indicates a slightly different positioning of the peptide in our AlphaFold model. However, the PYTV threonine hydroxyl group remains within hydrogen-bonding distance of the catalytic D74, supporting the plausibility of the modeled interaction.

The slight shift in peptide positioning observed in the MtPMT-PYTV complex may be attributed to the presence of the additional C-terminal valine (V) in PYTV, which was required by AlphaFold 3 due to its minimum ligand length constraint (≥ 4 amino acids). Additional modeling attempts using N-terminally extended peptides such as GPYT or PPYT resulted in significant displacement of the peptide, with the threonine positioned far downstream from D74 and buried deeply in the cavity normally occupied by the lipid donor.

Furthermore, the lipid molecule observed in the ScPMT1 protomer of the Bai et al. complex (previously shown in Figure 3c) has been omitted from the updated representation, as it clashes sterically with the terminal valine in the AlphaFold 3 model of the MtPMT-PYTV complex.

Legend modification: c. Close-up view of the AlphaFold 3 model of the PYTV peptide (shown as colored sticks) bound within the active site of MtPMT (cyan cartoon) (ipTM = 0.61, pTM = 0.92), superposed on the cryo-EM structure of ScPMT2 (grey cartoon) bound to a PYT peptide (grey lines; PDB: 6P2R, Bai et al.). The structural alignment (PyMOL RMSD_{MtPMT-PYTV / ScPMT2-PYT} = 2.39 Å) suggests a slightly altered peptide orientation in MtPMT, yet the threonine hydroxyl of PYTV remains within hydrogen-bonding distance of the catalytic D74, supporting the plausibility of the modeled interaction. d.

Similarly, Figures 4b and 4c have been revised to illustrate the modeled binding position of the [PPVPTTAASP] peptide (derived from the Apa1 sequence used in nanoDSF experiments) within the

aromatic cradle on the periplasmic face of MtPMT, as predicted by AlphaFold 3 (ipTM = 0.69, pTM = 0.88).

Legend modification: b. Surface representation of the periplasmic face of the MtPMT^{AF3} complexed with the Nter truncated Apa1 decapeptide [PPVPTTAASP] (ipTM = 0.69, pTM = 0.88), illustrating the substrate-binding groove. The [PPVPTTAASP] peptide is represented with carbon and nitrogen atoms in blue and oxygen in red. Key structural elements are highlighted: EL1Nter (yellow), EL4Cter (orange), the aromatic cradle (green), and the catalytic residue D74 (red). c. Cartoon representation of the putative acceptor peptide binding site in MtPMT, highlighting residues that form the aromatic cradle. Coloring is consistent with panel b.

Line 273: While I think this experiment is really interesting, and shows that these compounds can rescue the mutation, your data does not actually show that they “were able to diffuse through the bacterial envelope, [and] reach the mutated enzyme at the membrane.” You haven’t shown active engagement of these compounds with the protein directly. While less likely, some other series of events may be leading to this complementation. I would temper this statement to something closer to ‘Interestingly, we found that either guanidine chloride or imidazole were able to restore FasC mannosylation to approximately...’

We fully agree with this overinterpretation of this experience and have therefore modified the text as suggested:

« Interestingly, we found that either guanidine chloride or imidazole were able to ~~diffuse through the bacterial envelope, reach the mutated enzyme at the membrane, and restore FasC mannosylation to approximately 25 ± 5% of the level of the wild type MtPMT [54]~~ (Fig 3d). “

Line 387 – I am not sure P396 is the residue you intend to discuss here? It appears to be nowhere near the other two in the predicted structure.

Yes, we are indeed referring to residue P396 at this point in the discussion. Although it is spatially distant from the other two residues, P359 and W362, we believe it may correspond to the invariant C-terminal proline characteristic of WW domain signatures (PROSITE entry PS01159), as defined by O. Staub et al. (1996; DOI: 10.1016/S0969-2126(96)00054-8) and L. Otte et al. (2003; DOI: 10.1110/ps.023320). Our data show that this proline, located downstream of the second β -strand (with a pLDDT >90), is essential for MtPMT activity. This suggests it may play a key role in anchoring and stabilizing the β -sheet secondary structure of the EL4 loop.

Minor comments:

Line 193 – analysis, not analyzes

Modified

Line 200 – ‘key’ seems redundant with ‘selected’ here, I’d use one or the other

Modified

Line 219 – in order to -> ‘in order’ can be deleted and lose no meaning while improving readability

Suppressed as suggested

Line 259 – odd line break

Suppressed

Line 264 – I wouldn't quite call lysine and arginine 'chemically equivalent' I would just delete that part of the sentence

The text « the conservative substitution of R441 with a chemically equivalent basic lysine (R441K) almost completely preserved enzymatic activity,” has been modified as follow: “, the conservative substitution of R441 with a basic lysine (R441K) almost completely preserved enzymatic activity”

Line 377 – ‘functional folding’ is awkward, not totally sure what that means

Replaced by “position”

Figure 3a – the grey text (clone b exp2) in the inset is hard to read at this size. When typeset for publication I think it will be unreadable.

The figure has been modified accordingly

Figure 3c and elsewhere – It is far easier to understand the context of residues/ligands if they are coloured by heteroatom with the residue carbons a sufficiently contrasting colour. I would not colour the lipid carbons orange as you have, as this is typically used in phosphates.

The figure has been revised based on the comments, and the lipid has been removed for the reason outlined in the response to comment 1

Supplemental: There should be a table with quality metrics for the predicted structures where available. With AF models it is simple to color by pLDDT or report the average pLDDT. Admittedly the standards of reporting on models is only starting to grow, but the readers should have some sense of how much confidence the modelling software itself has in the prediction.

Quality metrics of the models of the MtPMT complexes are now indicated in the figures legends, while the model and pLDDT values of the AF3 model of the Mtb Rv1002c gene product are reported in Figure S1a together with the quality metrics (RMSD) of the AF3 model overlap with the different models obtained with other software.

Reviewer #2 (Remarks to the Author):

The discovery that protein O-mannosylation (POM) also occurs in bacteria, particularly in clinically significant mycobacteria, highlights its role as a potential virulence factor (see key reference 9 in the current manuscript). This finding has spurred interest in the characterization of the corresponding enzyme, MtPMT. The appeal of MtPMT is strengthened by its structural distinction from eukaryotic PMTs, which are heterodimeric complexes. In contrast, MtPMT, encoded by a single gene, is likely to function as a monomeric protomer.

Against this background, the present study aims to elucidate MtPMT's membrane topology and structure-function relationships while evaluating its viability as a drug target. Using a comprehensive suite of bioinformatics tools, the authors predict MtPMT's membrane topology and functional sites, validating some theoretical predictions through transfection experiments in a PMT-depleted *M. smegmatis* strain.

The membrane topology experiments, designed with a well-structured set of fusion constructs, are solid and convincing and so are cellular experiments controlling for functional expression and expression levels.

We thank the referee for the detailed evaluation of our work. We appreciate the recognition of the importance of protein O-mannosylation (POM) in mycobacteria and the potential of MtPMT as a unique bacterial target, as well as the acknowledgment of the strength and rigor of our membrane topology experiments and functional expression controls.

However, investigations into the predicted mechanistic elements and target selection features (WW-motif based recognition of Ser/Thr residue in proline-rich protein segments) lack unequivocal evidence. Additionally, some aspects of these experiments suffer from insufficient controls.

Before addressing specific weaknesses, I summarize this study as interesting but, in its current form, not meeting the standards for a high-impact publication. The proposed structure-function relationships and target selectivity studies remain largely speculative. More critically, the study does not address the key questions that arise throughout the text — questions the authors themselves succinctly outline in the final paragraph of the discussion (page 26, line 610 ff). Here a copy of the relevant passage: "(1) whether MtPMT dimerizes like its eukaryotic counterparts, (2) how bacterial PMTs might compensate for the absence of MIR domains, (3) whether MtPMT physically associates with the Sec complex as a formal component of the translocon machinery, and finally (4) how the acceptor sequence of the target protein is translocated, reaches the periplasm, and arrives at the reaction site of 618 membrane-embedded MtPMT." Significant information to these questions is essential before MtPMT can be seriously considered as a drug target.

We fully agree that the structural and mechanistic characterization of MtPMT is still in its early stages and that several critical questions—explicitly listed in the final paragraph of our discussion (page 26, line 610 ff.)—remain unanswered. As correctly noted, these include the potential for dimerization, compensation for the absence of MIR domains, interaction with the Sec translocon, and the mechanism of substrate trafficking to the active site. These questions define our future research directions, and we are currently designing experiments to address them.

However, we believe that our current findings lay essential groundwork for these future studies and support the rationale for considering MtPMT in early drug discovery efforts. Continued efforts to address the open mechanistic questions will be more particularly crucial for refining the search-space for chemical inhibitor identification and advancing the therapeutic potential of targeting bacterial O-mannosylation.

Major points:

The authors claim to have demonstrated the "druggability" of MtPMT by compensating for the loss of the guanidinium side chain of R441 in a mutant using a guanidinium salt buffer. While I agree that this in cellulose experiment (showing 25% reconstitution) suggests accessibility of the active site pocket, the study lacks

experiments that probe the functional consequences of interfering with the WW-motif, claimed to be essential for MtPMT activity.

With respect to our guanidinium rescue experiment in the R441A mutant, we respectfully clarify that we do not present this as definitive proof of druggability. Rather, we interpret the partial restoration of function *in vivo* as evidence that the active site is chemically accessible—an important first step in establishing that MtPMT can be modulated by small molecules *in vivo* and an essential prerequisite for any future therapeutic development. In response to the concerns raised by referees 1 and 2, we have revised the manuscript to present this interpretation more cautiously. Importantly, we also emphasize that the guanidinium rescue specifically targets the R441 residue in the EL5 loop, which is hypothesized to contribute to donor substrate positioning. This experiment does not address or involve any interference of the chemical with the EL4 WW-like domain.

Although this element may distinguish MtPMT from host PMTs, the WW-motif is abundant in the host, and proline-rich motifs are common in mammalian proteins. These motifs play critical roles in protein-protein interactions, influencing signaling pathways, structural stabilization, and the regulation of O-glycosylation and phosphorylation. All together massively challenging the evaluation of MtPMT as a drug target.

Indeed, the widespread presence of WW domains in eukaryotic proteins involved in key processes such as phospho-signaling and the regulation of metabolic pathways represents a major challenge for the development of inhibitors targeting these domains. Nonetheless despite extensive efforts to identify pharmacologically relevant WW domain inhibitors, no generic inhibitor has yet been developed for these highly degenerate structures. The conservation of WW domains is primarily based on a three-stranded β -sheet secondary structure and the presence of five conserved residues [P-W-Y-W-P] within a 20–35 amino acid segment. These residues define the characteristic WW domain signature, which is known to mediate interactions with proline-rich sequences.

However, the molecular basis for the selectivity of WW domains toward distinct proline-rich motifs remains poorly understood. Currently, WW domains are classified into four groups based on the types of proline-rich sequences they recognize. Nonetheless, even within a single group, individual WW domains exhibit different substrate specificities, which are further modulated by additional structural elements of the host protein. We address this complexity in the manuscript (lines 565, 599 and following).

This structural and functional diversity among WW domains and their substrates supports the idea that it may be feasible to selectively inhibit WW domains in therapeutically relevant proteins without causing systemic effects on all WW domain-containing proteins. This concept underlies many ongoing efforts to develop highly specific WW domain inhibitors with therapeutic potential—such as Pin1 in certain cancers.

The distinct cellular context of bacterial and eucaryote's PMTs'—embedded in a distinct membrane architecture and potentially mediating unique substrate recognition mechanisms— and the structural and selectivity differences (discussed in line 601) between the “WW-like domain” of bacterial MtPMT and the equivalent domains of PMT/POMT or canonical eukaryotic WW domains suggest that it should be possible to identify selective MtPMT inhibitors that do not interfere with human POMTs.

We therefore fully agree with the reviewer's primary comment noting that these “structural distinctions” between the bacterial enzyme and its human homologs strengthen the appeal of MtPMT” as a potential drug target”.

Furthermore, in the mutational experiments conducted on the C-terminus of EL4 to probe interactions with potential ligands, it would be preferable to mutate not only one interaction partner but also confirm the findings by introducing mutations in the second partner. Taken together, the experiments described on page 18 ff. are interesting but require proper controls. The authors appear to be aware of these issues, as much of the text is framed in a tentative manner.

We have tentatively addressed this point by testing distinct relevant peptides of the bacterial PMT protein substrate *in vivo*. We agree that a more detailed structural and biochemical study *in vitro* based on reciprocal punctual modification of interacting partners would be valuable to further characterize the structural elements involved in the interaction and strengthen the interpretation. However, the essential validation of such *in vitro* interaction data through mirroring *in vivo* experiments remains a substantial and challenging task, primarily due to the intricate and poorly resolved glycosylation patterns of FasC, the natural substrate of the bacterial PMT *in vivo*.

That's the reason why, while our results support a role for this region in acceptor substrate interaction, we refrain from making definitive mechanistic claims at this stage.

Minor

Page 3, line 63: It seems the cited literature is not adequate. Instead of 9,18 the citations 19,20 seem to be more fitting. In general the allocation of literature should be carefully controlled.

Reference 18, which details this process in eukaryotes, predates references 19 and 20 and is cited here for its foundational role. Similarly, reference 9 represents the first study to establish a parallel mechanism in prokaryotes by identifying the enzyme responsible for saccharide motif elongation. These references were therefore selected at this point in the text to highlight the chronological and conceptual progression in both systems.

Mix-up of numbering in Fig. S4

Modified

6: subnumbering is missing

Revised

ER does not need explanation but there are other truncates where the reader would profit from explanation.

Done

Minor grammar and spelling errors throughout the text

Reviewer #3 (Remarks to the Author):

Geraud and colleagues perform a thorough structure-guided study to confirm the predicted topology of MtPMT and to functionally validate key residues for *M. tuberculosis* PMT activity. Although they do not solve the protein structure, which, due to its nature, is an extremely difficult endeavour, they use high-quality models and structural comparisons with orthologues and homologues to infer the structure of this protein and identify residues to explore. The authors performed an impressive amount of work to address these questions, including mutagenesis on active site residues and residues involved in peptide substrate recognition, which is the highlight of this work.

The identification of a WW-like domain is novel and compelling, although the absence of structural validation is a limitation. My only concern is the exclusive reliance on DSF as the biophysical validation method. This aspect would benefit from additional validation, and it is unclear whether the authors tested other biophysical methods. This limitation may reflect the low affinity of these interactions, making them challenging to measure. However, the lack of further biophysical validation is not fundamental for the publication of this work if it cannot be obtained, but attempts should be made to explore alternative methods.

We thank the reviewer for its thorough and encouraging evaluation of our manuscript. We are grateful for the positive assessment of our structure-guided approach and functional analyses, particularly the

recognition of our mutagenesis work and the identification of a WW-like domain which we agree represents a novel finding.

Concerning the reviewer's comment on the exclusive use of differential scanning fluorimetry (DSF) for biophysical validation, we fully acknowledge this limitation and appreciate its suggestion to explore additional techniques. In response, we would like to clarify that, in addition to DSF, we also explored other techniques during the course of the study, including isothermal titration calorimetry (ITC) and proton nuclear magnetic resonance ($^1\text{H-NMR}$). While $^1\text{H-NMR}$ proved technically unfeasible due to sensitivity constraints, and ITC yielded preliminary but ultimately limited results.

As the reviewer correctly noted, the low affinity and transient nature of the interactions between synthetic EL4 peptides and their ligands posed significant challenges. These characteristics generally prevented the detection of binding by ITC under our experimental conditions, with the notable exception of the Apa1 ligand, which is derived from the natural Apa substrate of MtPMT. Consequently, we initially chose not to include these data in the original manuscript.

Although the reviewer has indicated that additional biophysical validation is not essential for publication, we now propose to include the ITC data as supplementary material in the revised manuscript.

Figure S5c: Isothermal titration calorimetry (ITC) analysis of the interaction at 37°C between the Apa1 ligand and the synthetic EL4Cter* peptide (a, b; ratio peptide/ligand: 33/700), or its W10A mutant analogue (W10A)EL4Cter* (c, d; ratio peptide/ligand: 25/288)

We believe these results, while limited, offer a more comprehensive perspective and further support our DSF findings. Under our experimental conditions, DSF proved to be the most robust and sensitive method for detecting ligand-induced thermal stabilization, enabling more detailed and interpretable analyses.

We hope the additional data and clarifications address the reviewer's concern and strengthen the overall experimental rigor of our study.

The manuscript is very well-written throughout, and references are appropriate. Figure 4b would benefit from a closer view of the EL4 and EL1 regions and higher resolution.

Figure 4b has been modified as suggested

Additionally, Line 193 "HR-MS analyzes" should be corrected to "HR-MS analysis."

Modified

Apart from these minor points, I have no further suggestions or criticisms of what is a thoughtful and well-executed work.